# Marine-Derived Actinomycetes: Biodegradation of Plastics and Formation of PHA Bioplastics—A Circular Bioeconomy Approach

**DOI:** 10.3390/md20120760

**Published:** 2022-12-01

**Authors:** Juliana Oliveira, Pedro L. Almeida, Rita G. Sobral, Nídia D. Lourenço, Susana P. Gaudêncio

**Affiliations:** 1Associate Laboratory i4HB, Institute for Health and Bioeconomy, NOVA School of Science and Technology, NOVA University Lisbon, 2819-516 Caparica, Portugal; 2UCIBIO-Applied Molecular Biosciences Unit, Chemistry and Life Science Departments, NOVA School of Science and Technology, NOVA University Lisbon, 2819-516 Caparica, Portugal; 3I3N-CENIMAT, Materials Science Department, School of Science and Technology, NOVA University Lisbon, 2819-516 Caparica, Portugal; 4Physics Department, Instituto Superior de Engenharia de Lisboa, Instituto Politécnico de Lisboa, 1959-007 Lisbon, Portugal

**Keywords:** (micro)plastics pollution challenge, actinobacteria, polymer biodegradation, biodegradation quantification methods, polyhydroxyalkanoates (PHAs), degradable bioplastic formation, blue biotechnology, circular economy

## Abstract

Plastics are present in the majority of daily-use products worldwide. Due to society’s production and consumption patterns, plastics are accumulating in the environment, causing global pollution issues and intergenerational impacts. Our work aims to contribute to the development of solutions and sustainable methods to mitigate this pressing problem, focusing on the ability of marine-derived actinomycetes to accelerate plastics biodegradation and produce polyhydroxyalkanoates (PHAs), which are biodegradable bioplastics. The thin plastic films’ biodegradation was monitored by weight loss, changes in the surface chemical structure (Infra-Red spectroscopy FTIR-ATR), and by mechanical properties (tensile strength tests). Thirty-six marine-derived actinomycete strains were screened for their plastic biodegradability potential. Among these, *Streptomyces gougerotti*, *Micromonospora matsumotoense*, and *Nocardiopsis prasina* revealed ability to degrade plastic films—low-density polyethylene (LDPE), polystyrene (PS) and polylactic acid (PLA) in varying conditions, namely upon the addition of yeast extract to the culture media and the use of UV pre-treated thin plastic films. Enhanced biodegradation by these bacteria was observed in both cases. *S. gougerotti* degraded 0.56% of LDPE films treated with UV radiation and 0.67% of PS films when inoculated with yeast extract. Additionally, *N. prasina* degraded 1.27% of PLA films when these were treated with UV radiation, and yeast extract was added to the culture medium. The main and most frequent differences observed in FTIR-ATR spectra during biodegradation occurred at 1740 cm^−1^, indicating the formation of carbonyl groups and an increase in the intensity of the bands, which indicates oxidation. Young Modulus decreased by 30% on average. In addition, *S. gougerotti* and *M. matsumotoense*, besides biodegrading conventional plastics (LDPE and PS), were also able to use these as a carbon source to produce degradable PHA bioplastics in a circular economy concept.

## 1. Introduction

Plastics are very useful materials due to their remarkable thermo-elastic and mechanical properties, which confer high resistance, stability and durability, chemical inertness, malleability, low water permeability, light weight, and low cost. For these reasons, plastics are produced and applied on a large scale [1]. However, the current production and consumption patterns implemented in society have harmful impacts on the environment [2]. LDPE and PS are among the principal marketed plastics worldwide, reaching values of 17.5% and 7.8% of total plastic consumption, respectively [3]. Due to the lack of proper waste management and their low rate of degradation, plastics have been accumulating in the environment, mostly in landfills and marine and coastal environments [1,4]. A total count of 525 × 10^10^ plastic particles is estimated to exist in the ocean globally [5].

Once in the environment, plastics begin to lose their integrity [6], firstly due to physical and chemical environmental factors, the so–called abiotic degradation caused by UV radiation, erosion, waves, and wind, among others, and subsequently, plastics lose their mechanical integrity through biological factors, named biotic degradation, caused by the action of microorganisms [7], which results in the fragmentation of the material into progressively smaller particles (<5 mm), termed microplastics [7,8,9]. Microplastics are considered environmental contaminants and emerging pollutants, threatening the health of organisms, including humans, and having negative ecological, social, and economic impacts [10,11]. It is, therefore, imperative to find solutions and sustainable methods to mitigate the global problem of plastic pollution. 

Plastic biodegradation has been investigated with the aim of mitigating plastic pollution. Biodegradation occurs through the action of bacteria, fungi, and yeast, which produce enzymes that cleave the plastics’ polymeric chains [12]. In particular, actinomycetes produce hydrolytic enzymes that allow them to grow on different polymers by facilitating the degradation of high molecular weight compounds into simpler ones [13]. For instance, the genera *Rhodococcus* and *Streptomyces* have been associated with plastics biodegradation [14,15]. The extent of biodegradation is usually observed by dry weight reduction and by morphological and chemical changes in their surface, among others [14,15,16,17,18]. 

In order to diminish the use of conventional plastics, bioplastics such as polylactic acid (PLA) have been proposed and are considered as an option to reduce the problem of plastic waste disposal through an alternative end-of-life scenario [19]. PLA is produced by the fermentation of biomass on feedstocks, such as sucrose, corn, and tapioca starches [20]. However, PLA is not readily biodegradable since it is only recyclable or biodegradable under industrial composting conditions (i.e., the mixed microbial community in aerobic conditions, with relative humidity > 65% and temperature > 55 °C) [19]. Several studies reporting the biodegradation of PLA have been published [21,22,23] and described *Amycolatopsis* as the dominant PLA-degrading actinomycete genus [24].

PHAs are biodegradable bioplastics polymers that are capable of replacing conventional plastics since these are industrially and home compostable, biodegradable in several environments, and anaerobically digestible, decreasing the environmental impacts of their use. PHAs are easily biodegradable by various microorganisms that produce extracellular PHA depolymerases [25,26] and can be totally degraded within 28 days in the sea or freshwater environments [27]. 

This study aims to contribute to the development of solutions and sustainable methods to mitigate the described problem by discovering new plastics biodegrading actinomycetes, using an available inhouse collection [28,29] located at UCIBIO, FCT-NOVA, Blue Biotechnology and Biomedicine Lab, to evaluate their potential to biodegrade different plastic polymers, developing a new approach for plastics biodegradation, and simultaneously, evaluating the ability of actinomycetes to produce PHA, using LDPE, PS and PLA plastics as carbon source, and thus assessing a novel circular economy approach. 

## 2. Results and Discussion

In the search for plastics pollution solutions, we consider that the development of novel biodegradation processes and the production of biodegradable bioplastics are the most promising mitigation approaches to plastics [9,30], and that the most efficient way to address this pollution challenge is to replace conventional plastics by eco-friendly bioplastics before they get into the oceans. Microorganisms and their metabolic activity can be used as an alternative strategy to common methods of recycling conventional plastics in a circular economy approach. 

### 2.1. Evaluation of Plastics Biodegradation Potential by Actinomycetes Using Plastics Emulsified Media Assays

The actinomycetes evaluated in this work were isolated from Estremadura Spur pockmarks and were taxonomically identified at the species level by sequencing of the 16S rRNA gene in a previous study [30]. Of the thirty-six screened actinomycetes strains, fourteen showed the ability to grow in plastics emulsified media (PVDF, PS, and PLA) after two weeks of incubation. The three actinomycete strains with higher potential to degrade plastics were identified by the formation of a clear zone halo: *Streptomyces gougerotti* PTE-001, *Micromonospora matsumotoense* PTE-025, and *Nocardiopsis prasina* PTE-048 (Figure 1). 

*S. gougerotti* grew extensively in PS emulsified medium, and the halo around the colonies measured 4 mm. *M. matsumotoense* grew on PVDF emulsified media forming a halo with a diameter of 1 mm; this strain also grew in PS emulsified medium but did not show a clear zone halo. *N. prasina* grew extensively in PLA emulsified medium after only seven days, forming a clear halo of 1 mm in half the time of the other two strains. 

Three strains, *S. gougerotti*, *M. matsumotoense*, and *N. prasina,* demonstrated a higher capability of degrading conventional plastics (PVDF, PS, And PLA) than the remaining 33 actimomycete strains that were used in these plastics emulsified media biodegradation assays. Thus, these strains were selected to perform subsequent thin plastic films biodegradation assays in liquid media. 

In our study, *Streptomyces* and *Micromonospora* were the most dominant genera regarding plastic degradation. Other reported species belonging to the *Streptomyces* genus, *Streptomyces* KU8, *S. badius*, *S. setonii*, and *S. viridosporus*, have been described as capable of biodegrading different polymer types, especially polyethylene (PE) [17,31,32]. The *Micromonospora* genus has never been previously reported for plastics biodegradation. Moreover, there are no studies reporting PLA degradation by the *Nocardiopsis* genus [9,33].

### 2.2. Evaluation of Plastics Biodegradation by Actinomycetes Using Thin Plastic Films Assays

Thin plastic films were used in this assay in order to uniformize the shape of the plastics and to allow an easier evaluation of the biodegradation at the film’s surface both by weight loss, FTIR-ATR, and mechanical tests. According to the results of the previous biodegradation screening assays, *S. gougerotti* and *M. matsumotoense* were used for LDPE and PS films biodegradation assays and *N. prasina* for PLA film biodegradation assays.

Thin plastic films biodegradation assays were analyzed and quantified by weight loss, FTIR-ATR spectroscopy, and mechanical assays. The results obtained with these methods are used in a combined approach to conclude that biodegradation effectively occurred [34]. 

The biodegradation assays were performed for polymeric thin plastic films of LDPE, PS, and PLA. PVDF was substituted by LDPE pellets, as PVDF was only available as powder, which hampered its molding in a heated press. LDPE was not used for the preparation of an emulsified medium due to solubility issues. The PVDF backbone chain contains alternate C–F and C–H bonds, while PE only contains C–H bonds (Figure 2). Generally, polymers lose small molecular fragments of their side chains and the polymeric backbone by bond scission. In this case, during degradation, PVDF typically loses fluorine and hydrogen fluoride, forming new bonds, mainly C–C double bonds. During PE degradation, PE oligomers with C–C double bonds are formed, as well as carbonyl groups [35,36].

#### 2.2.1. Thin Plastic Films’ Weight Loss

Weight loss of the thin plastic films used in the biodegradation assays provided the first insights into the potential of the selected actinomycete strains to degrade LDPE, PS, and PLA films. Measurements of weight loss are frequently used when the polymers are exposed to selected microorganisms in culture media, with the polymers as the sole carbon source [37]. This method is standardized for in-field and simulation biodegradability assays [38]. 

The thin plastic films’ weight loss for each biodegradation assay is presented in Table 1. Weight loss was calculated and compared with the corresponding controls considering the same period, i.e., the assays were monitored for 60–180 days and were performed in triplicate.

LDPE is one of the major sources of environmental pollution since large quantities of this plastic are accumulated in the environment [17]. It is important to highlight that studies focusing on LDPE degradation by actinomycetes are very rare [14,16,39]. In the biodegradation assays, the LDPE films incubated with *S. gougerotti* and *M. matsumotoense* lost 0.5% and 0.31% of their weight, respectively, in 180 days, while the corresponding control did not lose any weight. Moreover, the LDPE films that were treated with UV radiation and inoculated with *S. gougerotti* lost 0.56% of their initial weight in 90 days, showing that UV pre-treatment accelerated *S. gougerotti* capacity to biodegrade LPDE.

PS is considered a non-biodegradable polymer, and it is not easily recycled. Furthermore, the biodegradation efficiency of PS is not as successful as the biodegradation of LDPE [40,41]. During the biodegradation assays, the regular PS films incubated with *S. gougerotti* and *M. matsumotoense* lost about 0.17% and 0.04% of their initial weight in 180 days, respectively. The obtained results confirmed that this polymer is extremely resistant to biodegradation. Additionally, the assays with PS films treated with UV radiation and inoculated with *S. gougerotti* and *M. matsumotoense* achieved a weight loss of 0.30% and 0.17% in 90 days, respectively, while the control films did not lose weight. 

In addition to the previous conditions, yeast extract was added to the culture medium as a nitrogen source to promote the growth of actinomycetes biomass to determine if this component enhances the biodegradation results. *S. gougerotti* exhibited the highest potential to degrade PS in the screening by the clear zone halo test. Consequently, PS films inoculated in this condition lost 0.67% of their initial weight. The PS UV films, for the same condition but for a shorter period (90 days), decreased by 0.19% of the initial plastic film weight. The weight of the control films remained unchanged. The presence of yeast extract in culture media resulted in an increase in weight loss of over 3-fold, from 0.17% to 0.67% of the initial weight in 180 days.

PLA is a biopolyester that is produced from renewable resources [21]. Nevertheless, PLA is not readily biodegradable unless under industrial composting conditions [19]. In total, 3 different PLA biodegradation assays were performed with *N. prasina* and monitored for 60 days. *N. prasina* reduced the weight of regular PLA films, PLA UV-treated films, and PLA UV-treated films in the presence of yeast extract in the culture media by 0.84%, 0.97%, and 1.27%, respectively. 

#### 2.2.2. Evaluation of Thin Plastic Films Chemical Changes by FTIR-ATR Spectroscopy 

The most relevant spectral differences observed for each thin plastic film (LDPE, PS, and PLA) after the biodegradation assays are described in Table 2. The changes in the chemical functional groups observed on the surface of the thin plastic films under study were compared to the corresponding controls using FTIR-ATR; Figure 3, Figure 4 and Figure 5 show examples of these chemical structural changes. Appendix A show the spectra for the remaining conditions.

The characteristic LDPE absorption bands were identified: 2915 cm^−1^ (CH_2_ asymmetric stretching), 2847 cm^−1^ (CH_2_ symmetric stretching), 1472 cm^−1^ and 1461 cm^−1^ (bending deforming), 730 cm^−1^ and 718 cm^−1^ (rocking deforming) [37,42,43]. The spectra of the LDPE films incubated with *S. gougerotti* (Figure 3a,c) or *M. matsumotoense* (Appendix A) showed three new bands compared to the corresponding control. A band at 1739 cm^−1^ (C=O stretching) corresponds to the formation of carbonyl groups, and bands at 1367 cm^−1^ and 1217 cm^−1^ (C–H), which together with a 1468 cm^−1^ band represents isopropyl groups that indicated a decrease in the degree of polymerization [37,43]. 

For the LDPE films treated with UV radiation incubated with *S. gougerotti* (Figure 3b,d), small intensity bands appeared at 1740 cm^−1^, corresponding to the carbonyl group formation by UV radiation exposure (abiotic degradation) and at 1600 cm^−1^, representing C–H stretching that along with 1468 cm^−1^ band represents isopropyl groups, indicating a decrease in the degree of polymerization [44]. 

For the LDPE films that were treated with UV radiation and incubated with *S. gougerotti* (Figure 3b,d) or *M. matsumotoense* (Appendix A), the main difference in both spectra was the appearance of a band at 1096 cm^−1^. This band indicates the presence of oxidized groups (groups containing –OH) resulting from bacterial biodegradation [45,46]. 

Ghatge, S. et al. (2020) reported a biodegradation mechanism by bacteria for PE [36] that was presented in Appendix A, in which carbonyl groups (–C=O) and oxidized groups such as moieties containing –OH groups are formed, as it was observed in our FTIR-ATR spectra.

Regarding the PS spectra, the characteristic bands occurred at 3059 cm^−1^ and 3025 cm^−1^ (aromatic C–H stretching vibration absorption), 2921 cm^−1^ and 2849 cm^−1^ (methylene –CH_2_ asymmetric and symmetric stretching), 1600 cm^−1^, 1492 cm^−1^, 1449 cm^−1^, and 1027 cm^−1^ (corresponding to deformational vibrations of both –CH_2_ and aromatic C=C stretching vibration absorption, which indicates the existence of benzene rings), 906 cm^−1^ (vinyl C–H out-of-plane a small bump band), 751 cm^−1^ (C–H out-of-plane bending vibration of the aromatic ring, which indicates that there is only one substituent in the benzene ring), 695 cm^−1^ (ring-bending vibration), and 538 cm^−1^ (alcohol OH out-of-plane bend) [40,47]. Furthermore, for PS films treated with UV radiation, the intensity of the bands increased, which can also be a sign of biodegradation. This increase was more noticeable at 695 cm^−1^ and evidenced a monosubstituted benzene ring in the polymer or PS depolymerization [48,49]. 

For the PS films that were incubated with *S. gougerotti,* a new band at 1739 cm^−1^ was observed in the FTIR-ATR spectrum, which indicates the formation of carbonyl (–C=O) groups during PS degradation (Figure 4a,b). Interestingly, at 3600 cm^−1^, a small bump can be noticed, which represents O–H stretching, suggesting that alcohols can be generated as an intermediate stage on the way to the carbonyl formation during PS degradation [50,51]. Moreover, new bands were detected between 1000 and 1200 cm^−1^ that correspond to carboxylic acids and esters, and some of the bands shifted to lower intensities, suggesting that the bonds of these chemical groups become weaker in the PS films [52]. Furthermore, the addition of yeast extract to the culture medium resulted in a general increase in the intensity of the bands (Appendix A). This information supported the evidence of the oxidation processes during PS biodegradation. A possible mechanism for PS biodegradation reported by Mooney, A. et al. (2006) is presented in (Appendix A). The bands observed in the FTIR-ATR spectra correspond to this reported mechanism [53], where bands corresponding to formed chemical groups in the polymer surface, such as carbonyl groups (–C=O), alcohols (–C–OH), carboxylic acids (–COOH), esters (–COOR), and bound alterations in the benzene rings.

In what concerns PS films incubated with *M. matsumotoense*, the intensity of the bands decreased over time (Appendix A). The decrease in the intensity of the bands indicates the biodegradation of the polymers by the secretion of enzymes from the microorganisms, which leads to polymeric disintegration. The changes in the intensity of the bands at 695 cm^−1^ and 755 cm^−1^ indicated that the chemical structure of PS was slightly transformed after the biodegradation assays due to depolymerization. Furthermore, a new band at 1735 cm^−1^ increased its intensity when compared to the corresponding control. This absorption band indicated the formation of carbonyl groups (–C=O) during the degradation of the studied polymer [54,55].

For PS films treated with UV radiation and incubated with *S. gougerotti*, a new band emerged in the FTIR-ATR spectrum at 1739 cm^−1^ (Figure 5a,b). At the same time, a general decrease in the intensity of the bands was observed when compared to the corresponding control time (Appendix A). For the same period, the films incubated with *M. matsumotoense* presented a general decrease in the intensity of the bands (Appendix A). Regarding PS UV-treated films incubated with *S. gougerotti* in culture media supplemented with yeast extract, the intensity of the bands increased, namely the bands at 1360 cm^−1^ and 1050 cm^−1^, which represent carboxylic acids and esters, and the band at 1739 cm^−1^ was no longer visible (Figure 5). This is evidence of thin plastic film oxidation; once in the earlier stages of biodegradation, carbonyl groups are formed as intermediate products, and when biodegradation advances, these groups are consumed by the bacteria [56]. 

The characteristic absorption bands of PLA are at 2996 cm^−1^ (C–H asymmetric stretching); 1749 cm^−1^ (stretching vibration of C=O group); 1453 cm^−1^ and 1365 am^−1^ (asymmetric and symmetric vibrations of CH_3_ group, respectively); 1180 cm^−1^, 1125 cm^−1^, 1082 cm^−1^, and 1044 cm^−1^ (stretching vibration of O–C–C); 867 cm^−1^ (vibration of the O–CH_2_–CH_3_ group); and at 753 cm^−1^ and 699 cm^−1^ (deforming vibration of the αCH_3_) [57]. The main observed difference between the FTIR spectra of PLA and UV-treated PLA was in the intensity of the absorption bands, which decreased with the photo-oxidation treatment (Appendix A).

The biodegradation assays with *N. prasina* performed with regular PLA films resulted in the overall decrease in the absorption bands when compared to the corresponding control (Appendix A), with the three absorption bands at ~1750 cm^−1^, ~1185 cm^−1^, and 1090 cm^−1^ the ones that were more related to biodegradation [58]. Additionally, the intensity of the absorption bands of UV-treated PLA films increased, and new bands appeared at 1382 cm^−1^, 1127 cm^−1^, and 704 cm^−1^ when compared to the corresponding control. These results indicate that chain oxidation occurred during biodegradation and induced the cleavage of the long chains into shorter ones. The band, at ~700 cm^−1^, corresponds to the formation of C–O double bonds [42,43]. Pattanasuttichonlakul, W. et al. (2018) reported a generic bacteria biodegradation mechanism for PLA that is presented in Appendix A [59], which reflects what was observed in the FTIR-ATR spectra obtained for PLA films after the biodegradation assays, such as biodegradation chain oxidation and formation of vinyl unsaturated groups.

Generally, the intensity of all the absorption bands of the UV-treated PLA films incubated with *N. prasina* in a culture medium supplemented with yeast extract increased when compared to the corresponding control (Appendix A). This increase was higher than the one observed for UV-treated PLA films without the yeast extract addition. Moreover, new bands were noticed at 1382 cm^−1^, 1266 cm^−1^, 1128 cm^−1^, 956 cm^−1^, and ~700 cm^−1^. The bands at ~950 cm^−1^ are characteristic of vinyl unsaturated groups, and the increase of around 700 cm^−1^ corresponds to C–O double bonds, which are related to biodegradation [60].

#### 2.2.3. Mechanical Properties of the Thin Plastic Films

Tensile strength assays were used to determine the mechanical properties of the various polymers studied in this work (LDPE, PS, and PLA). These tests provided a measurement of the ability of these plastics to resist forces that tend to deform them, determining to what extent these polymers can be deformed before breaking [61]. The degree of degradation of the polymers was determined by measuring the tensile strength of each thin plastic film, for which different parameters were analyzed, such as the Young Modulus (E), the Yield Strength, and the Ultimate Tensile Strength (UTS). The loss of tensile strength is a characteristic indication of biodegradation, as the cleavage of the polymer’s backbone during degradation weakens the material, so the tensile strength is proportional to the extent of degradation [62,63]. The tensile strength results are presented in Table 3.

The Young Modulus is a mechanical property that measures the tensile stiffness of solid materials and quantifies the relation between tensile stress (force per unit area) and axial strain (proportional deformation) in the linear elastic region of the polymers. The Yield Strength is defined as the stress threshold at which a specific amount of permanent deformation occurs, i.e., until this threshold value, the polymer can return to its original dimensions. The UTS is the maximum stress that a material can withstand while being stretched or pulled before breaking. For fragile materials, the UTS value is close to the Yield Point, while in ductile materials, this value can be higher [64].

In general, the standard Young Modulus for LDPE is between 110 and 450 MPa. The Young Modulus obtained for “standard” thin LDPE films and UV-treated LDPE films prepared in the laboratory was approximately 831 MPa and 618 MPa, respectively [65].

LDPE films incubated with *S. gougerotti* lost 31% of their Young Modulus and 30% of their Yield Strength during the biodegradation assays. Moreover, when LDPE films were incubated with *M. matsumotoense,* the Young Modulus decreased by 40%. The UTS values showed a lower decrease in both assays. The results of these assays are presented in Figure 6 as an example.

All the tested parameters using UV-treated LDPE films incubated with *S. gougerotti* decreased when compared to the corresponding control. In this case, the Young Modulus, the Yield Strength, and the UTS values decreased by 12.4%, 25.5%, and 5.4%, respectively. For the films incubated with *M. matsumotoense* a decrease of 31.8% was observed in the Young Modulus parameter (Table 3). 

The standard values of Young Modulus and UTS for PS are 3000–3500 MPa and 30–100 MPa, respectively [65]. The PS films produced in this study, before any assay, presented Young Modulus, Yield Strength, and UTS values of 1467.8 MPa, 12.6 MPa, and 21.1 MPa, respectively. The UV-treated PS films presented Young Modulus, Yield Strength, and UTS values of 818.6 MPa, 11.9 MPa, and 12.3 MPa, respectively. The differences between the UV non-treated and treated films are visible in the value variation of both the Young Modulus and the UTS, which can validate the success of UV irradiation of the films as a treatment to improve biodegradation.

The mechanical tests of the PS films incubated with *S. gougerotti* showed no decrease in the obtained parameters through the biodegradation assay. Moreover, PS films incubated with *M. matsumotoense* showed a 32% reduction of the Young Modulus values. 

In the case of PS films incubated with *S. gougerotti* in the culture medium supplemented with yeast extract, a significant variation in the Young Modulus was observed; this parameter decreased by approximately 59%. Regarding the UV-treated PS films, for the same assay condition, the Young Modulus decreased its value by about 56%. 

The standard Young Modulus, Yield Strength, and UTS values for PLA are 2300 MPa, 35.9 MPa, and 26.4 MPa, respectively [66]. The values obtained for these parameters of PLA films prepared in the laboratory were 2554 MPa, 61 MPa, and 66.4 MPa. For UV-treated PLA films, the values obtained were 864 MPa, 28 MPa, and 29 MPa, respectively. All the parameter values decreased after the UV treatment, which indicates the efficiency of the UV treatment on PLA films.

Regarding PLA films incubated with *N. prasina* a significant decrease in the Young Modulus was observed when compared to the corresponding control. This parameter decreased by 18% during the assay. The Young Modulus of UV-treated PLA films incubated with and without yeast extract added to the culture media also decreased during the assays by 29% and 2%, respectively. According to Table 3, no variation in the Yield Strength and the UTS values were detected for this polymer under the used conditions.

### 2.3. PHA Inclusions in the Actinomycetes Cells

Members of different microbial genera biosynthesize polymeric lipids, such as PHAs, and about 300 PHA-producing bacterial strains have been identified [67]. PHA is produced by several bacteria species as an internal carbon storage and energy reserve under different conditions [67]. When essential nutrients, such as nitrogen and phosphate, are limited and the carbon source is in excess, bacteria with the potential to produce PHA accumulate these biopolymers as intracellular inclusions [68]. The thin plastic films biodegradation assays with actinomycetes were performed with an excess of carbon source and a limitation of other nutrients. The presence of PHA inclusions in the actinomycetes cells was assessed to evaluate the PHA production potential of the selected actinomycetes by using plastics as a carbon source. Thus, in the current study, the detection of PHA inclusions in the actinomycetes cells was used to determine the potential of these bacteria to produce biodegradable bioplastics using conventional plastics as their carbon source.

The three strains used in the different liquid media biodegradation assays were stained and visualized with an epifluorescence microscope to search for PHA inclusions. The strain *S. gougerotti,* when incubated with PS films, UV-treated PS films, and UV-treated LDPE films, revealed PHA vesicles with fluorescence intensity (Figure 7a–c, respectively).

It was possible to clearly observe PHA accumulation in *M. matsumotoense* when incubated with UV-treated PS films. PHA vesicles were not visible for the strain incubated with LDPE or UV-treated LDPE, or PS films. These results suggest that *M. matsumotoense* can use UV-treated PS as the sole carbon source since UV irradiation facilitates microbial attack and the beginning of biodegradation (Figure 8a,b).

### 2.4. LDPE Biodegradation and PHA Production by Actinomycetes

In the biodegradation assays performed with LDPE films treated and non-treated with UV radiation, the films treated with UV radiation were expected to deteriorate to a greater extent than non-treated films. Generally, the use of UV radiation is applied to obtain lower molecular weight molecules since UV radiation works as an initiator of PE oxidation. The polymeric carbon bonds are broken during this process, which leads to photo-induced degradation [69], producing carbonyl groups that promote the microorganisms’ attack [70]. In this way, a combination of photo-oxidation by UV radiation exposure and the use of microorganisms to perform plastics biodegradation should create a synergistic effect to improve plastics degradation. Yet, we observed that UV treatment did not have much impact on the weight loss of the polymeric LDPE films, and the films’ weight losses were not as high as expected, considering the studies already carried out with PE incubated with actinomycete species. For example, incubation of this polymer with *Rhodococcus ruber* resulted in a 7.5% weight loss in 60 days, in synthetic medium (NH_4_NO_3_ (1 g/L); K_2_HPO_4_ (1 g/L); MgSO_4_·7H_2_O (0.2 g/L); KCl (0.15 g/L); CaCl_2_·2H_2_O (0.1 g/L); and microelements: FeSO_4_·6H_2_O (1 mg/L), ZnSO_4_·7H_2_O (1 mg/L), MnSO_4_ (1 mg/L)), at 30 °C and 150 rpm [14]. For UV-treated LDPE films incubated with the same species, isolated from soil, an 8% weight loss occurred in 30 days, also in synthetic medium, at 30 °C and 150 rpm [71]. For UV-treated PE films, *Streptomyces albogriseolus* LBX-2, *Micrococcus* sp., and a bacterial consortium constituted by *Arthrobacter viscosus*, *M. lylae*, *M. luteus,* and *Bacillus* sp. were able to achieve significant percentages of weight loss, 17.3%, 6.6%, and 17%, respectively. All these bacteria were isolated from different soil samples. Gause’s medium (KNO_3_ (10 g/L), K_2_PO_4_ (5 g/L), MgSO_4_ (5 g/L), FeSO_4_ (0.1 g/L), and NaCl (5 g/L)) was used in *S. albogriseolus* LBX-2 biodegradation assay, but in the other two studies, the plastic films were buried in soil in contact with bacteria [39,72,73,74]. The different results of this study, when compared to the literature, can be due to differences in the small scale of the experiments (100 mL culture media) and acclimatization of the strain to LDPE. However, the extent of biodegradation cannot be based solely on the reduction of film weight. In fact, the results of FTIR-ATR spectroscopy and the mechanical assays revealed LDPE biodegradation. Regarding the FTIR-ATR spectra, new bands were observed at 1739 cm^−1^ when compared to the corresponding control, which indicates the formation of carbonyl groups (–C=O), an intermediate product of biodegradation. The emergence of this band is an indicator of biodegradation, already reported in the literature, and demonstrates that the strains were capable of biodegrading or oxidizing PE structures [50]. Additionally, the two bands that showed between 1200 and 1300 cm^−1^, which represent the oxidized groups, such as moieties containing –OH groups, are related to oxidized products formed during LDPE biodegradation [75]. Carbonyl groups are produced through the degradation stages, which enhance the biodegradation process. In this way, only the oxidized and short LDPE chains are metabolically used by the microorganisms. Ghatge, S. et al. (2020) reported a mechanism for PE biodegradation by bacteria that illustrates the mentioned formed groups (Appendix A) [36]. Usually, after a long incubation time and advanced oxidation, methylene units are cut-off by intracellular enzymes in β-oxidation and are completely degraded via the citric acid cycle resulting in the formation of CO_2_ and H_2_O [72,75,76]. For the LDPE films that were treated with UV radiation, a new absorption band was observed at 1096 cm^−1^, indicating the presence of oxidized groups formed by the activity of the microorganisms [45,77]. Considering the spectra results, *S. gougerotti* and *M. matsumotoense* revealed LDPE chemical changes due to degradation both before and after UV treatment. Therefore, both *S. gougerotti* and *M. matsumotoense* degraded LDPE after UV treatment. 

The results of the mechanical tests for LDPE and UV-treated LDPE films showed that the Young Modulus was the parameter that decreased the most in both biodegradation assays. The Young Modulus measures the necessary load to induce a given deformation in the polymer. As the Young Modulus decreases over time, the load needed to induce deformation also decreases, being a biodegradation indicator. LDPE is a long-branched polymer, so as polymer chains are broken during biodegradation, most probably the branches are degraded first by the action of extracellular enzymes produced by the actinomycetes strains. This can explain the higher decrease in the Young Modulus parameter than in the other two tested parameters. Therefore, the differences in Young Modulus support the evidence of LDPE biodegradation by *Streptomyces gougerotti* and *Micromonospora matsumotoense*. The fact that the films became fragile was an indication of the preliminary stages of microbial decomposition [76]. Other studies reported a reduction in the percentage of other parameters (elongation, for example) in mechanical assays after the biodegradation process [72,76,78,79]. One reported study also showed a reduction in the percentage of elongation at break when LDPE was inoculated with a *Streptomyces* species [80].

PHA vesicles were observed when *Streptomyces gougerotti* was incubated with UV-treated LDPE films. To the best of our knowledge, no studies using marine-derived actinomycetes have been performed both for conventional plastics biodegradation and for their use as a carbon source for PHA formation. In fact, this is the first study that relates marine-derived actinomycetes with PHA. In previously reported studies using soil actinomycetes the potential of PHA production has been evaluated by genotypic and phenotypic studies, namely by PCR detection and sequencing analysis of the *pha*C genes that encode the PHA synthase [81]. Some *Streptomyces* strains have been described as harboring these *pha*C genes, although *S. gougerotti* still needs further investigation. The most relevant actinomycetes genera in terms of PHA production are *Streptomyces*, *Rhodococcus*, *Nocardia*, and *Corynebacterium* [68,82]. In this work, marine-derived *S. gougerotti* has demonstrated to have potential to produce PHA using conventional plastics as a carbon source. 

### 2.5. PS Biodegradation and PHA Production by Actinomycetes

In what concerns the weight loss of the PS films incubated with *S. gougerotti* or *M. matsumotoense*, there were differences between the weight loss of UV-treated and non-treated PS films. In this case, the UV radiation treatment improved the effect of biodegradation, resulting in higher weight losses, which is in agreement with prior investigations [83]. In the literature, PS biodegradation studies performed with actinomycetes were not as successful as the ones for LDPE. For example, *Rhodococcus ruber*, isolated from soil, only reduced the dry weight of this polymer by 0.8% in 60 days, while in the same period, the same species could degrade LDPE dry weight by 7.1%. The growth conditions were practically the same (synthetic medium 120 rpm, and 35 °C) [14,84]. However, there are studies with higher PS weight loss percentages. Therefore, the use of microorganisms with the potential to degrade PS may be a viable solution for reducing the accumulation of this plastic in the environment, although further studies for yield optimization are needed.

*Streptomyces gougerotti* exhibited the highest potential to degrade PS in our screening. Thus, to test if different media conditions could enhance biodegradation by this strain, yeast extract was added to the culture medium. It is described that supplementation with this component can improve biodegradation [85]. Yeast extract is an excellent organic source of nitrogen, with high content of proteins and amino acids, vitamins, minerals, and growth factors that influence the growth and type of secondary products formed by *Streptomyces*, increasing mycelium biosynthesis [85,86]. In our study, UV radiation and the addition of yeast extract showed to increase the percentages of weight loss. Therefore, when comparing thin PS films incubated in a culture medium without and with yeast extract, weight losses increased from 0.17% to 0.67%, respectively. Furthermore, when UV-treated PS films were incubated without yeast extract for 90 days, *S. gougerotti* only degraded 0.09% of the polymer weight, but when this strain was incubated with yeast extract for the same period of time, it degraded 0.19%, twice the percentage of weight loss. Studies using *Streptomyces* genera in biodegradation assays with yeast extract in the culture medium demonstrated a reduction in the integrity of polymers which sustained the benefit of the addition of this component [80]. Other studies that also used yeast extract in the culture media reported an improvement in PS biodegradation when compared to the ones without yeast extract [87]. 

The FTIR-ATR spectra of PS films incubated with *S. gougerotti* and *M. matsumotoense* showed differences when compared with the corresponding control. In these cases, bands corresponding to the formation of carbonyl groups were observed, as well as a band corresponding to the generation of alcohols, which support the evidence of the biodegradation potential of these two strains. The obtained results for UV-treated PS films showed the potential of *S. gougerotti* and *M. matsumotoense* to alter the PS surface and initiate the first steps of biodegradation. Generally, irradiation with UV light promotes the degradation of PS macromolecules, leading to the reduction of the molecular weight and acceleration of the biodegradation rate [83]. PS is constituted by styrene monomers, and styrene itself can be used as a carbon source by some microorganisms [84]. The use of UV irradiation treatment in other biodegradation studies was particularly efficient for *Rhizopus oryzae, Aspergillus terreus*, and *Phanerochaete chrysosporium* strains, as it enhanced the biodegradation by these species. In these cases, an increase in the band intensities at 536 cm^−1^, 748 cm^−1^, 1026 cm^−1^, 1361 cm^−1^, 1450 cm^−1^ (C=C stretching vibration of aromatic compounds), 1735 cm^−1^, and 3313 cm^−1^ occurred in only 60 days [48]. After the addition of yeast extract to the culture media, major differences were observed for the UV-treated PS films. In this case, the band corresponding to the carbonyl groups could not be detected, which may evidence oxidation of the films since, as biodegradation advances, the carbonyl groups that were formed are consumed by the microorganisms. The obtained results evidenced *S. gougerotti* potential to degrade PS in a medium supplemented with yeast extract. Thus, when the strains were grown in conditions close to optimal, plastic degradation was enhanced. FTIR-ATR results revealed that PS biodegradation differs with the microorganism and microbial communities, highlighting the complex process beyond biodegradation. The variations in the intensities of the functional groups at the surface of the polymers are due to their consumption or formation [7].

The difficulty of PS biodegradation was evidenced by the mechanical tests. In other studies on PS, higher variations in mechanical properties occurred when the polymer was blended with bioplastics since this type of mixture accelerates molecular structural changes [88,89]. Plastics/bioplastics blends can be a possible solution for the future production of products made of PS, as it enhances biodegradation, which will be beneficial after disposal [89]. The addition of yeast extract to the culture medium in the assays with *S. gougerotti* resulted in significant differences. As mentioned before, the addition of yeast extract improves the reduction of the integrity of polymers and thus promotes a more efficient PS degradation [80,87]. Thus, it should lead to the weakening of the mechanical properties of PS. The obtained results for the Young Modulus evidenced the potential of *S. gougerotti* to degrade PS. Statistical significance was found for Yield Strength and UTS values (*p*-value < 0.05) when yeast extract was added to the culture medium. This indicates that the addition of yeast extract enhances the biodegradation of thin plastic films since these two parameters are associated with the degradation of polymeric chains.

PHA inclusions were observed in *S. gougerotti* by epifluorescence microscopy after incubation with PS and UV-treated PS films (Figure 7). These inclusions were also clearly observed for *M. matsumotoense* incubated with UV-treated PS films, showing the potential of *M. matsumotoense* to use pre-treated PS as the sole carbon source (Figure 8). Our results are in accordance with the literature, as soil *Streptomyces* species are already known for their capacity to produce PHA. However, no information exists about the presence of the phaC gene in the *Micromonospora* genus, but there is evidence for the accumulation of triacylglycerols [90]. Considering this information, other species of the *Micromonospora* genus may have the potential to produce PHA, such as *M. matsumotoense*. Actually, actinomycetes have been recognized as PHA producers using glucose as a carbon source, specially *Rhodococcus*, *Nocardia*, and *Streptomyces* species [68]. Specifically, *Streptomyces toxytricini* D2 strain, isolated from polluted soil, produced under optimized conditions (8% tapioca molasses as carbon source, 8% of inoculum, 4% (NH_4_)_2_SO_4_, pH 6.5, at 30 °C), 23.64 g/L of the PHA granules were identified by ^1^H NMR, ^13^C NMR, and FTIR. This corresponds to 86.56% yield production of PHA, making *S. toxytricini* D2 an efficient candidate for the mass production of PHAs [91]. 

Other studies reported that *Streptomyces* isolates can produce relatively high amounts of biodegradable polymers. *Streptomyces subrutilus*, isolated from Colombian soil, produced 27.5% PHA in dry weight, also demonstrating good productivity in terms of biodegradable biopolymer production. The authors achieve this high productivity by using conditions for optimal growth, such as media containing glucose (5 g/L) as a carbon source, peptone (5 g/L) and yeast extract (3 g/L) at 25 °C, and 250 rpm [92]. Moreover, Alvarez and co-workers verified that *Nocardia corallina* and *Rhodococcus ruber*, isolated from soil samples from Brazil, produced significant amounts of PHA when the strains were grown in nitrogen starvation (0.26 g/L (NH_4_)_2_SO_4_) and media containing glucose/glucose and casein as carbon source and 5.65 g/L K_2_HPO_4_·3H_2_O, 2.38 g/L KH_2_PO_4_, and 1 g/L MgSO_4_·7H_2_O with 1 mL of micronutrient solution containing 6.4 g/L CuSO_4_·5H_2_O, 1.1 g/L FeSO_4_·7H_2_O, 7.9 g/L MnCl_2_·4H_2_O, and 1.5 g/L ZnSO_4_·7H_2_O [93]. 

These studies support the interest and highlight the potential of marine-derived actinomycetes to be used in the future production of biodegradable bioplastics. In this work, actinomycetes used plastic polymers as a carbon source to produce PHA, being, to the best of our knowledge, the first study to use this type of carbon source without thermal treatment for biodegradable biopolymers production [94,95]. Thus, the type of PHA that was produced and which sequence is being expressed by phaC gene should be further investigated, as different bacteria species have different phaC gene sequences [68]. Additionally, by optimizing the culture conditions, it will be possible to advance and develop future bioplastics production by actinomycete strains that are capable of producing polymeric lipids, such as PHA. To achieve this goal at the laboratory scale, fermentation optimizations, including temperature, pH, agitation, NaCl concentration, and media optimizations, can be performed to promote PHA formation [27]. NaCl Concentration plays an important role in the fermentation optimization of marine-derived bacteria [96,97,98].

### 2.6. PLA Biodegradation and PHA Production by Actinomycetes

The weight loss values obtained for PLA films are in accordance with the results obtained for the other tested plastics polymers, i.e., films without any treatment have lower percentages of weight loss when compared with thin UV-treated films and incubated in culture medium supplemented with yeast extract, for the same period of time. A comparison of the results obtained for PLA with LDPE and PS biodegradation assays for 60 days showed that PLA is more biodegradable than LDPE and PS. Reported studies proved that optimal conditions (polymeric film pre-treatment, yeast extract addition, etc.) for the biodegradation assays significantly enhanced the percentages of weight loss [85], and this was also observed in this study. According to Jarerat et al. (2002), PLA-degrading actinomycetes belong to the *Pseudonocardiaceae* family and related genera, which includes *Amycolatopsis*, *Saccharothrix*, *Lentzea*, *Kibdelosporangium*, *Actinomadura*, *Nonomuraea*, *Thermonospora*, *Thermopolyspora* and *Streptoalloteichus* [21,24,99,100]. In fact, several studies indicate that actinomycetes demonstrate an efficient PLA degradation capacity, degrading this biopolymer either under field trials or under laboratory conditions [24]. Jarerat and co-workers showed an efficient PLA degrading activity of a PLA-degrading enzyme produced by *Amycolatopsis orientalis*. The use of a purified enzyme in this study, produced by an actinomycete, allowed the total degradation of PLA at 30 °C and 140 rpm [100]. The same authors also studied the capacity of *Saccharothrix waywayandensis* to degrade PLA. The strain degraded 95% of PLA when the basal media contained 0.1% gelatine, at 30 °C and 150 rpm, for 3 months [21]. Members of the *Nocardiopsis* genus exhibit a vast metabolic versatility and are biotechnologically important, as many species of this genus can mediate the breakdown of complex polymers that occur naturally [101]. Our results indicate that *N. prasina* is a good target for polymer biodegradation investigation.

The FTIR-ATR results for PLA evidenced the occurrence of biodegradation on UV-treated PLA films and that supplementation of the culture medium with yeast extract improved this process. There are no reports in the literature regarding PLA degradation by members of the *Nocardiopsis* genus. In this way, *N. prasina* is a newly identified bacterial species with the potential to degrade this polymer. Biodegradation was confirmed in this and other studies by FTIR spectra analysis, which showed changes in the chemical structure of PLA, observed by the presence of new chemical functional groups vibrations, which indicate polymer biodegradation [57].

Regarding the mechanical tests, significant differences were found between UV-treated PLA films and regular films (*p*-value < 0.05) in the Young Modulus, which indicates that the UV treatment was effective and increased the biodegradation effects. 

Combining these results with the ones from the FTIR-ATR spectra, alterations were observed in the surface of the PLA films by the formation of C–O double bonds and unsaturated vinyl groups. With these changes, the polymer would become more resistant to deformational forces and stretching, which could explain the absence of variation in yield strength and UTS [102]. In reported studies, decreases in PLA strength are used to determine biodegradation when the films are incubated with microorganisms [103,104]. 

The strain *N. prasina* incubated with PLA and UV-treated PLA films presented autofluorescence by epifluorescence microscopy; thus, the presence of PHA inclusions was inconclusive. No PHA production studies with this marine-derived species have been reported before. 

## 3. Materials and Methods

### 3.1. Preparation of Polymer-Emulsified Media

Briefly, 1% of PVDF emulsified medium was prepared as follows: 0.05 g of PVDF powder was dissolved in 2.5 mL of dichloromethane, with the addition of 0.01 g of sodium lauryl sulphate (SLS) surfactant and 37.5 mL of filtrated natural seawater (NSW) and 12.5 mL of distilled water (proportion of 75:25%, respectively). The mixture was sonicated, using an ultrasonicator (Emmi-D60 from EMAG Technologies, Salach, Germany), for 15 min, at room temperature. After sonication, the dichloromethane was evaporated at 80 °C for 10 min with stirring. Before autoclaving, 0.7 g of agar was added to the solution [105,106].

Then, 1% of PS and PLA emulsified media were prepared as follows: 0.05 g of polymer pellets were dissolved in 3 mL of dichloromethane overnight. After the dissolution, 0.01 g of SLS surfactant, 37.5 mL of filtrated natural seawater (NSW), and 12.5 mL of distilled water were added and sonicated for 10 min at room temperature. After sonication, dichloromethane was evaporated at 80 °C with stirring for 10 min. Simultaneously, a minimum medium containing 37.5 mL of NSW, 12.5 mL of distilled water, and 0.7 g of agar was prepared and autoclaved to be used as a control. The prepared emulsions and the minimum medium, after being autoclaved, were poured into Petri dishes [106]. 

### 3.2. Actinomycetes for Plastics Biodegradation Screening

Marine-derived actinomycete strains were previously isolated from sediments collected off the Estremadura Spur (85 strains), on the coast of Continental Portugal, between *Carvoeiro* Cape and *Roca* Cape, and taxonomically characterized through 16S rRNA gene sequencing [29]. One representative strain of each different species retrieved from this ocean location, i. e. a total of thirty-six strains, was screened for plastics biodegradation using emulsified media of three different polymers (PVDF, PS, and PLA) in triplicate. After this assay, three strains were selected for thin plastic films biodegradation assays in liquid media (LDPE, PS, and PLA), *Streptomyces gougerotti* PTE-001, *Micromonospora matsumotoense* PTE-025, and *Nocardiopsis prasina* PTE-048. The GenBank accession number of the representative 16S rRNA gene sequence of each of these species is NR_041201.1, NR_025015.1, and NR_044906.1, respectively. 

### 3.3. Clear Zone Test for Plastics Biodegradation

Actinomycetes with the potential to degrade plastics were identified by the clear zone test in agar plates with a medium containing the polymer as a carbon source. Polymer degradation was analyzed after 2 weeks of incubation at 25 °C. The actinomycetes with the potential to degrade the plastics polymers were selected by their capacity to grow on the medium and by the formation of a clear zone halo around the colonies. These assays were performed in triplicate.

The size of the halo was measured, and the photos were taken using an optical microscope [85]. The selected actinomycetes, mentioned in 3.2, were considered potential polymer-degrading strains and used for the thin plastic films biodegradation studies.

### 3.4. Production and Preparation of Polymeric Thin Plastic Films 

PS, LDPE, and PLA pellets were transformed into thin films (with a thickness of approximately 100 µm) using a heated hydraulic press. To obtain thin films with smooth surfaces, aluminium plates with a clean surface were used between the heated press plates. Before the production of each film, it was necessary to thoroughly clean the aluminium metal plates and heat the press to the desired temperature. The temperature was selected considering the melting temperature (T_m_) of each polymer. For each film processing, it was required that the temperature of the press was higher than the T_m_ of each polymer. For LDPE, PS, and PLA, the T_m_ was 135 °C, 240 °C, and 180 °C, respectively [107,108]. Thus, the temperature used on the heated press was approximately 150 °C, 260 °C, and 200 °C, respectively (Table 4). In total, 1 g of LDPE and PS pellets were placed on the bottom of the aluminium plate until melting, i.e., until these became transparent. The other aluminium plate was then placed on top of the first one, and the polymer was pressed for 5 min, with a pressure between 10 and 20 bar. Finally, before the removal of the thin film, the aluminium plates were allowed to cool down to room temperature. After the separation of the two plates and when the polymer was fully cooled, the film was removed, and it was ready to be used in the biodegradation assays. For PLA, the process was similar, 1.5 g of PLA pellets were used, and the aluminium metal plates were replaced by aluminium foils.

After the production of the polymeric thin plastic films, half of the LDPE-, PS-, and PLA-prepared films were treated using UV light at Wavelength λ = 254 nm (Elnorlux, 3907RVT815, 15 W, 230 V, and 50 Hz) by direct exposure during 30 min, and stored at room temperature for further use [18]. 

Before being used in the biodegradation assays, the films were cut into small rectangles (5 × 20 mm) and cleaned with 2% SDS for 30 min, rinsed with distilled water, and then allowed to dry overnight at 50 °C. After drying, the films were placed in 70% (*v*/*v*) ethanol for 30 min and allowed to air dry in a laminar flow hood. The films were aseptically added to a sterilized Erlenmeyer [37,85].

### 3.5. Thin Plastic Films Biodegradation Assays

As previously mentioned, the strains that showed the highest potential to degrade plastics in the biodegradation screening were assessed by the growth and development of a halo and were selected for a biodegradation assay. 

A pre-inoculum was performed in a liquid medium (20 mL) and allowed to grow for 2 weeks before being used in the thin plastic films’ biodegradation assays. The culture medium was composed of 75:25% SSW:distilled water. In addition, another condition was tested by adding yeast extract (0.1% (m/v)) to the culture medium (20 mL); when using this condition, the biomass increased about 2-fold. Approximately 10% of the previous pre-inoculum biomass for each of the selected actinomycete strains was transferred into fresh medium (75:25%, SSW:distilled water) (100 mL) with a sterilized volumetric pipet. The sterile thin plastic films (LPDE, PS, and PLA) were added to these culture media and were incubated in an orbital shaker at 30 °C and 100 rpm. The control experiments were carried out in the same conditions, using a non-inoculated growth medium.

The biodegradation assays were performed with both regular thin plastic films and thin films pre-treated with UV radiation. The assays were performed in triplicate for 60–180 days. 

### 3.6. Monitoring of Thin Plastic Films Biodegradation

To monitor the plastics biodegradation process, the thin plastic films were periodically removed from the culture with sterile forceps. The formed bacterial biofilm was washed off by agitation in 2% (*w*/*v*) SDS for 5 h and then washed with distilled water. The films were placed on filter paper and dried overnight at 50 °C [14,37].

The biodegradation of plastic films was analyzed by comparing the weight loss and surface chemical structural changes by FTIR-ATR and mechanical assays.

#### 3.6.1. Plastic Films Weight Loss

The polymeric films were weighed on an analytical weighing scale accurate to 0.0001 g (OHAUS, Analytical Plus 110S, Nänikon, Switzerland) before incubation with the actinomycete culture and after being removed from the biodegradation assay and cleaned. The percentage weight loss was calculated by comparing the dry weight of the films with the initial value and calculated according to the following equation [85,109]:(1)$$\mathrm{Weight}\;\mathrm{loss}\;\left(\%\right)=\frac{m_{\mathrm{ini}}-m_{\mathrm{dry}}}{m_{\mathrm{ini}}}\times 100\%$$
where *m*_ini_ is the initial weight of the polymeric film before the degradation assay and *m*_dry_ is the dry weight of the polymeric film after the biodegradation assay. The weight loss was expressed in percentage (%). This assay was performed in triplicate.

#### 3.6.2. Fourier Transform Infrared Spectra of Plastic Films 

FTIR spectra of the film samples were recorded in the 400–4000 cm^−1^ range using a Perkin Elmer Spectrum Two Fourier Transform Infrared Spectrophotometer with attenuated total reflectance (FTIR-ATR). This method was used to analyze the chemical variations in the structure of the plastic films resulting from the biodegradation assays in comparison with the original polymers and the corresponding controls [57]. In the spectra, the y axis is Transmittance (T%), and the x axis is Wave Number (cm^−1^). These assays were performed in triplicate.

#### 3.6.3. Tensile Strength Tests

To test the differences in the mechanical properties of the films before and after the biodegradation assays, tensile tests were conducted on the films (5 × 20 mm) at room temperature, using uniaxial tensile testing equipment from Rheometric Scientific equipped with a load cell of 100 N. Film thicknesses (e) were measured with a digital micrometer with 1 μm resolution. Mechanical assays were performed by fixing the film ends in the claws of the tensile test equipment; the thin film was then deformed at a constant speed of 0.5 mm/min in opposite directions while the load was continuously registered. The load (N) and stroke (m) values were obtained for each test and were treated using Excel. The load (N) and stroke (m) values were converted to stress (*σ*) and strain (*ε*) using the following equations, respectively, for tests normalization: (2)$$\sigma =\frac{F}{A_{0}}\;\left[\mathrm{Pa}\right]$$
(3)$$\varepsilon =\frac{\Delta l}{l_{0}}\;\left[\mathrm{adimentional}\right]$$
where *F* is the load (N), *A*_0_ is the initial value of the area (m^2^) of the cross-section of each sample, Δ*l* is the stroke (m), and *l*_0_ is the initial length (m) between claws, i.e., the initial length of the sample. Based on these parameters, the Young Modulus (MPa), the Yield Strength (MPa), and the Ultimate Tensile Strength (UTS) (MPa) were determined. These assays were performed in triplicate.

### 3.7. Evaluation of Bioplastic Production by Detection of PHA Inclusions in Actinomycete Cells

To evaluate the production of PHA by the selected actinomycete strains, a method developed by Ostle and Holt (1982) and adapted from Serafim et al. (2002) was applied. A drop of Nile blue stain was added to ~1 mL of bacterial culture, which was then incubated at 55 °C for 15 min. After this incubation period, the sample was examined under an Olympus BX51 epifluorescence microscope (Tokyo, Japan) at 1000×. With this method, intracellular PHA granules were observed as bright orange fluorescence [110,111]. 

### 3.8. Statistical Analysis

Statistical analysis was performed with GraphPad Prim 5. The statistical significance was calculated and used to test the effects of UV radiation pre-treatment and the addition of yeast extract to the culture medium, using a two-tailed paired *t*-test with a 95% confidence interval [37,112].

## 4. Conclusions

Pollution caused by plastics and microplastics has serious negative impacts, including on the marine environment, health, society, and economy, which demands an urgent investigation for solutions to mitigate this problem. In this study, actinomycetes were used to biodegrade several types of plastic polymers (LDPE, PS, and PLA) and to use these plastics as a carbon source for the production of PHA biodegradable bioplastics. Herein, three novel actinomycete strains were identified as having the capability to biodegrade plastics: *Streptomyces gougerotti*, *Micromonospora matsumotoense*, and *Nocardiopsis prasina.* Biodegradation became more evident when the thin plastic films were artificially aged by UV exposure and when yeast extract was added to the culture medium. The production of PHA inclusions by *S. gougerotti* and *M. matsumotoense* was observed when these actinomycetes were incubated with PS, UV-treated PS, and UV-treated LDPE films for the first strain and UV-treated PS films for the second strain, respectively. Overall, this study highlights the potential of using microorganisms, namely marine-derived actinomycetes, in a circular economy manner to upcycle common plastics by their use as a carbon source to form biodegradable plastics, such as PHA. Moreover, this work paves the way for the future optimization of the fermentation processes for yield improvement so that it can contribute to mitigating plastic pollution issues. Currently, in our laboratory, fermentation optimizations, scale-up processes, and PHA identification are being implemented. 

## Figures and Tables

**Figure 1 marinedrugs-20-00760-f001:**
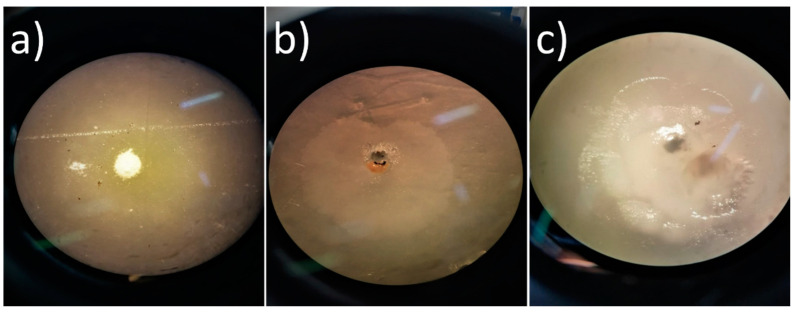
Plastics biodegradation by marine-derived actinomycetes (optical microscopy with (40×) amplification): (**a**) *Streptomyces gougerotti* clear zone halo formed in PS emulsified medium (4 mm), after two weeks of incubation (17 days); (**b**) *Micromonospora matsumotoense* clear zone halo formed in PVDF emulsified medium (1 mm), after two weeks of incubation (14 days); (**c**) *Nocardiopsis prasina* and the clear zone halo formed in PLA emulsified medium (1 mm), after one week of incubation (7 days). Each media contained 0.1% (*w*/*v*) of emulsified plastic polymer and agar in 75:25% natural seawater (NSW):distilled water. The biodegradation assays were performed in triplicate.

**Figure 2 marinedrugs-20-00760-f002:**
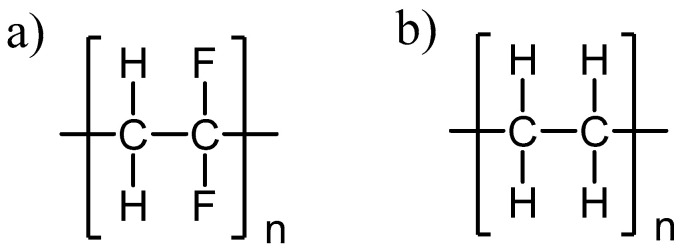
Polymers’ chemical structures: (**a**) PVDF and (**b**) PE.

**Figure 3 marinedrugs-20-00760-f003:**
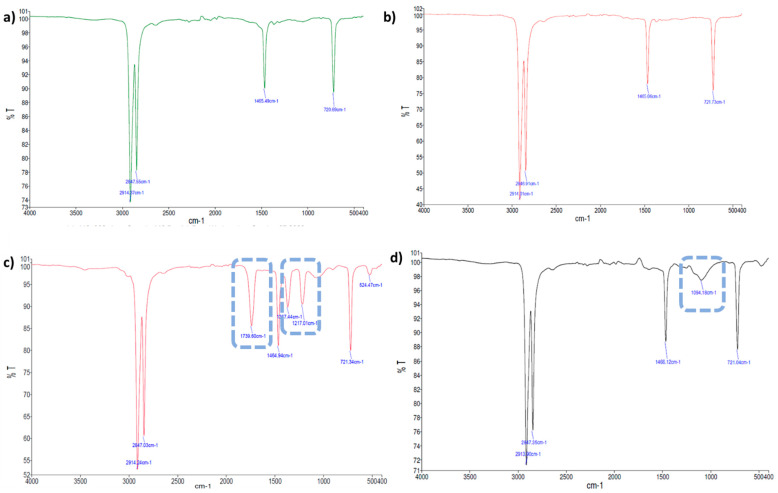
FTIR-ATR spectra of LPDE with *S. gougerotti* in culture medium: (**a**) LDPE films control; (**b**) LDPE incubated with *S. gougerotti*; (**c**) LDPE UV films control; and (**d**) LDPE UV films incubated with *S. gougerotti*. The dashed rectangles correspond to the new bands observed after incubation. In the FTIR-ATR spectra, *y* axis represents Transmittance T (%) and *x* axis Wave Number (cm^−1^).

**Figure 4 marinedrugs-20-00760-f004:**
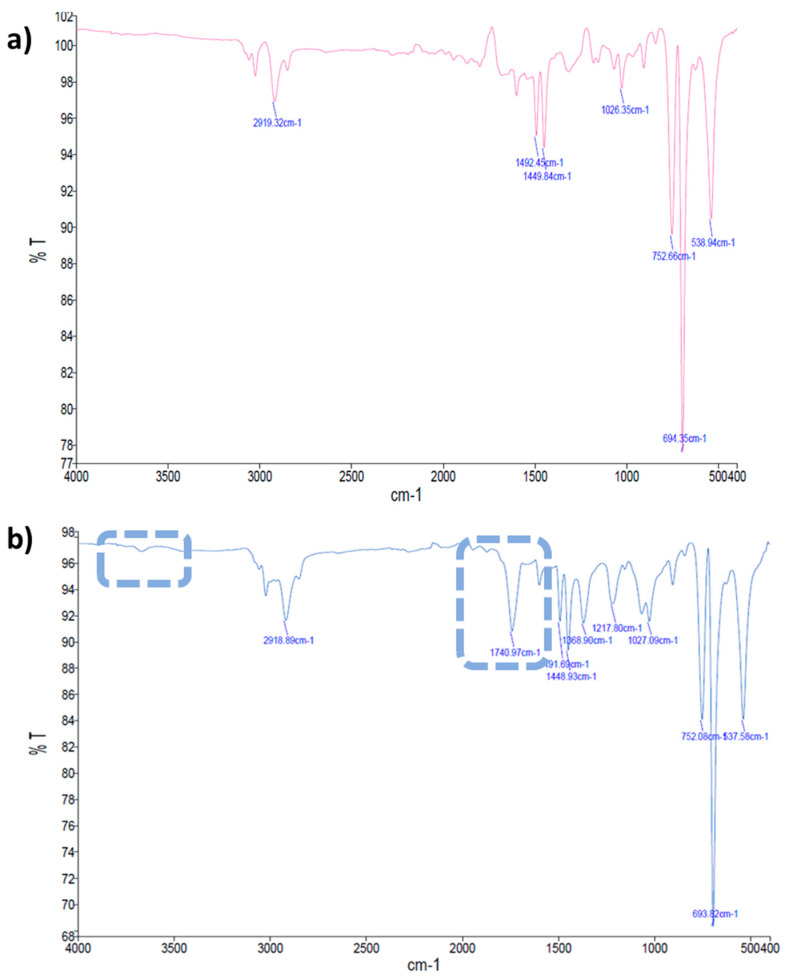
FTIR-ATR spectra of PS films incubated with *S. gougerotti* in culture medium. (**a**) PS control films, and (**b**) PS films that were films incubated with *S. gougerotti*. The dashed rectangles correspond to the newly observed bands after incubation. In the spectra, *y* axis represents Transmittance T (%) and *x* axis Wave Number (cm^−1^).

**Figure 5 marinedrugs-20-00760-f005:**
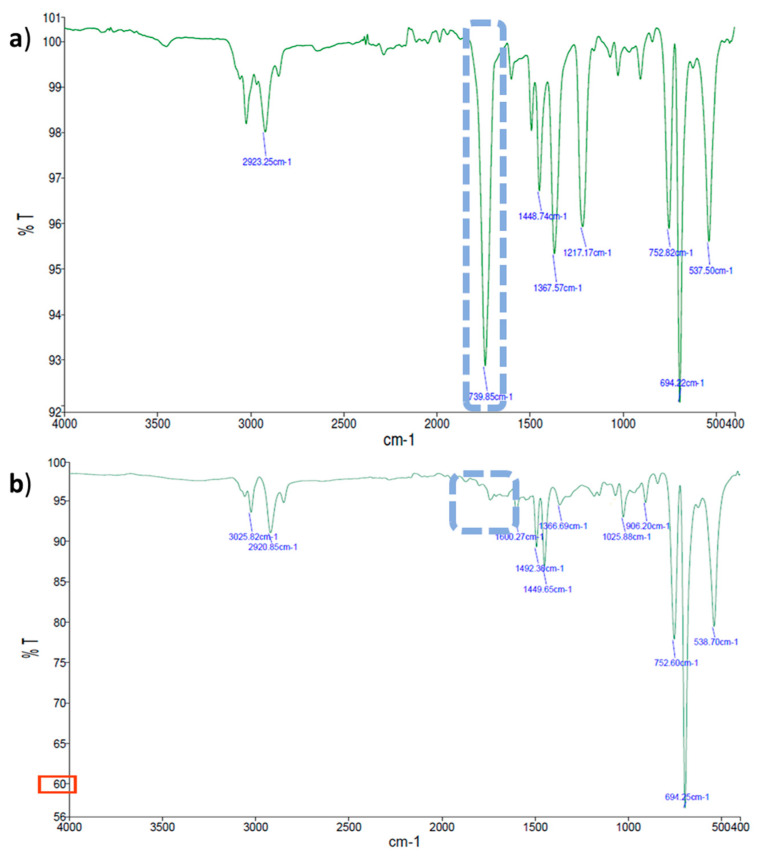
FTIR-ATR spectra of PS UV-treated films incubated with *S. gougerotti* in culture medium. (**a**) without yeast extract supplementation and (**b**) with yeast extract supplementation. The dashed rectangles correspond to the new band that was observed after incubation without yeast extract, and that was not observed upon incubation with yeast extract. Red rectangle highlights the increased intensity of the bands with condition (**b**). In the spectra, *y* axis represents Transmittance T (%) and *x* axis Wave Number (cm^−1^).

**Figure 6 marinedrugs-20-00760-f006:**
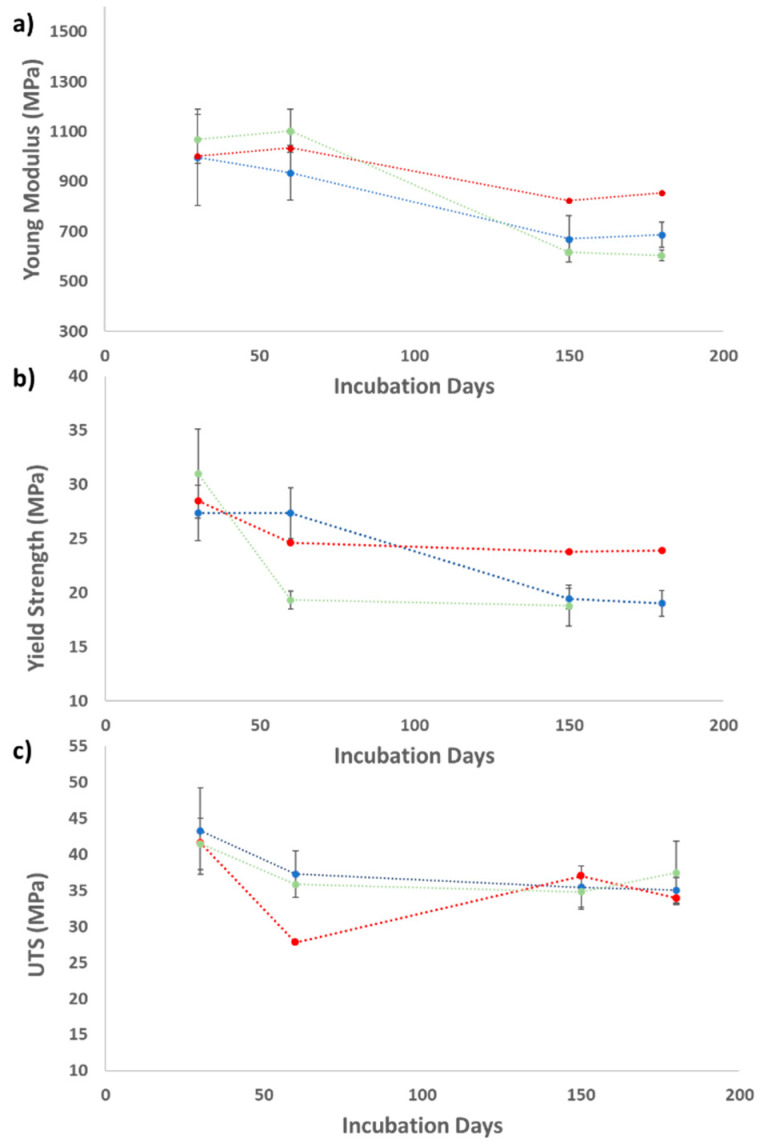
Tensile tests results for biodegradation assays performed with LDPE films. LDPE films were incubated with *Streptomyces gougerotti* and *Micromonospora matsumotoense*. (**a**) Young Modulus values; (**b**) Yield Strength values; and (**c**) UTS values (red circles—control film; blue circles—incubated film with *S. gougerotti*; green circles—incubated film with *M. matsumotoense*). A reduction of Young Modulus due to actinomycetes biodegradation was observed.

**Figure 7 marinedrugs-20-00760-f007:**
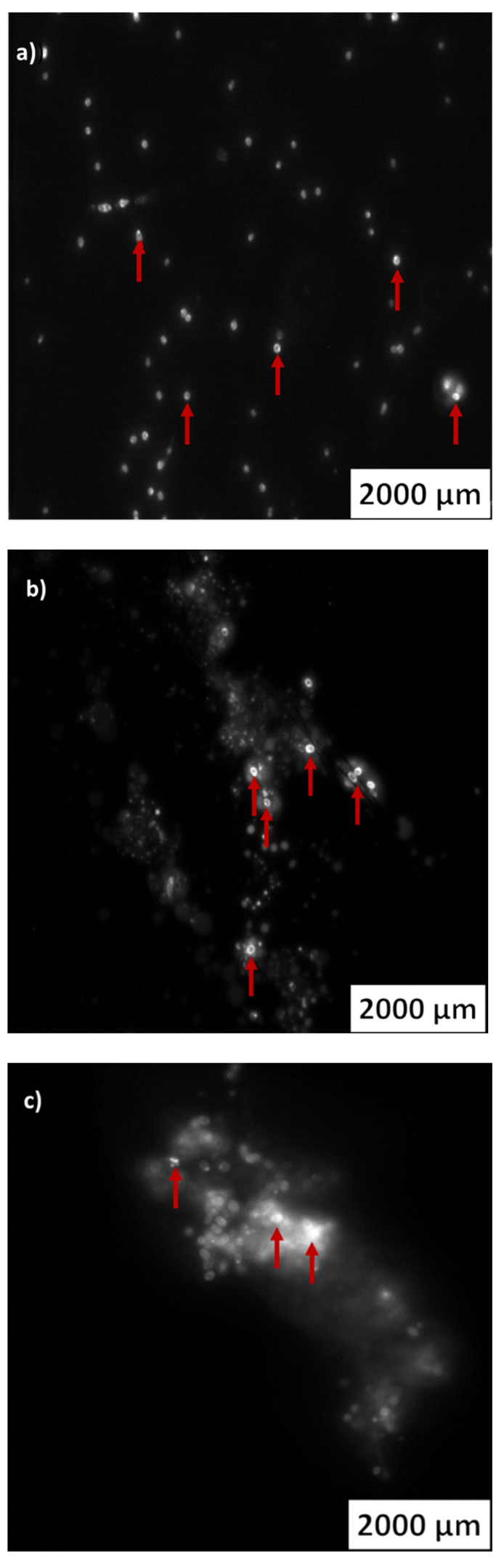
*S. gougerotti* PHA inclusions observed by epifluorescence microscopy: (**a**) incubation with PS films, (**b**) incubation with UV-treated PS films and (**c**) incubation with UV-treated LDPE films.

**Figure 8 marinedrugs-20-00760-f008:**
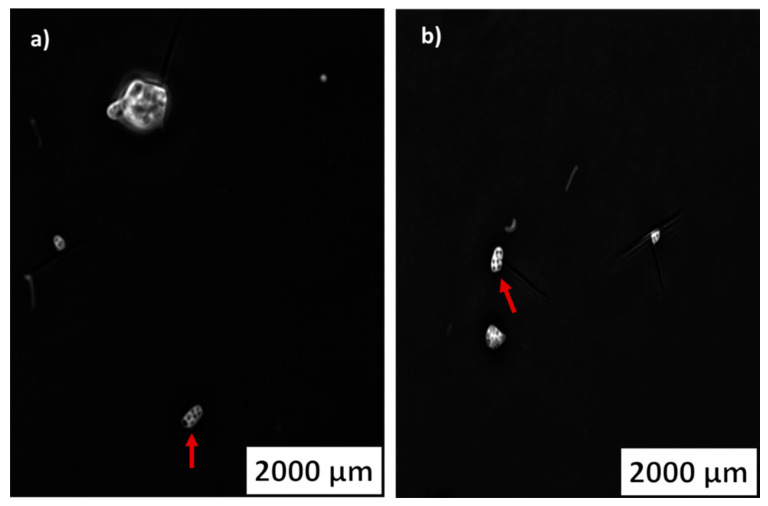
*M. matsumotoense* PHA inclusions observed by epifluorescence microscopy when inoculated with UV-treated PS films. Both (**a**,**b**) represent different *M. matsumotoense* cells that accumulated PHA.

**Table 1 marinedrugs-20-00760-t001:** Thin plastic films (LDPE, PS, and PLA) weight loss at the end of the biodegradation assays for each polymer type, selected strain, and culture medium (i.e., culture media with and without yeast extract).

Polymer Type	UV Treatment	Time (Days)	Actinomycete Strains	Weight Loss (%)
LDPE	No	180	*S. gougerotti*	0.50 ± 0.08
			*M. matsumotoense*	0.31 ± 0.02
	Yes	90	*S. gougerotti*	0.56 ± 0.12
			*M. matsumotoense*	0.22 ± 0.01
PS	No	180	*S. gougerotti*	0.17 ± 0.01
			*M. matsumotoense*	0.04 ± 0.00
	Yes	90	*S. gougerotti*	0.30 ± 0.11
			*M. matsumotoense*	0.17 ± 0.04
PS with yeast extract	No	180	*S. gougerotti*	0.67 ± 0.03
	Yes	90		0.19 ± 0.04
PLA	No	60	*N. prasina*	0.64 ± 0.20
	Yes			0.97 ± 0.07
PLA with yeast extract	Yes	60	*N. prasina*	1.27 ± 0.19

**Table 2 marinedrugs-20-00760-t002:** FTIR-ATR spectral chemical structural changes of the thin plastics films (LDPE, PS, and PLA) after the biodegradation assays.

Actinomycete Strains	Polymer Type	UV Treatment	Band Wave Number	Chemical Changes
*Streptomyces* *gougerotti*	LDPE	No	New bands at 1739, 1367, and 1217 cm^−1^	Formation of carbonyl groups; primary and secondary alcohols
		Yes	New band at 1096 cm^−1^	Oxidized functional groups
	PS	No	New bands at 3600 and 1740 cm^−1^	Alcohols generation; formation of carbonyl groups
		Yes	New band at 1740 cm^−1^; Decrease band intensity at 1026, 748, and 694 cm^−1^	Indication of oxidation; mono-substituted benzene rings
	PS with yeast extract	No	New bands at 1740, 1216, 1367, and 3600 cm^−1^ (low intensity)	Formation of carbonyl groups; carboxylic acids, esters; and generation of alcohols
		Yes	Increased band intensity; disappearance of the band at 1740 cm^−1^	Evidence of film oxidation
*Micromonospora matsumotoense*	LDPE	No	New band at 1740 cm^−1^	Formation of carbonyl groups
		Yes	New band at 1070 cm^−1^	Chain oxidation
	PS	No	Decrease bands intensity	Biodegradation of polymer chains
		Yes	Increased bands intensity	Indication of slight transformation of the chemical structure
*Nocardiopsis prasina*	PLA	No	Decrease bands intensity	Biodegradation of polymer chains
		Yes	New bands at 1382, 1127, and 704 cm^−1^	Chain oxidation
	PLA with yeast extract	Yes	New bands at 1382, 1266, 1128, and 700 cm^−1^	Formation of vinyl un-saturated groups; and chain oxidation

**Table 3 marinedrugs-20-00760-t003:** Tensile strength decrease percentage at the end of the biodegradation assay for each thin plastic film, selected strain, and culture medium (i.e., culture media with and without yeast extract). N.D.—Not detected under the used conditions.

Polymer Type	UV Treatment	Time (Days)	Actinomycete Strains	Young Modulus Decrease (%)	Yield Strength Decrease (%)	Ultimate Tensile Strength Decrease (%)
LDPE	No	180	*S. gougerotti*	31.1	30.6	19
			*M. matsumotoense*	40.7	18.3	9.7
	Yes	90	*S. gougerotti*	12.4	25.5	5.4
			*M. matsumotoense*	12.4	25.9	20.7
PS	No	180	*S. gougerotti*	N.D.	N.D.	N.D.
			*M. matsumotoense*	31.8	N.D.	N.D.
	Yes	90	*S. gougerotti*	N.D.	16.7	0.7
			*M. matsumotoense*	3.51	N.D.	N.D.
PS with yeast extract	No	180	*S. gougerotti*	58.8	N.D.	N.D.
	Yes	90		56.4	0.2	3
PLA	No	60	*N. prasina*	18.1	N.D.	N.D.
	Yes			2.43	N.D.	N.D.
PLA with yeast extract	Yes	60	*N. prasina*	27.4	N.D.	N.D.

**Table 4 marinedrugs-20-00760-t004:** Polymer pellets, quantities, and temperatures used to produce thin plastic films on a heat press.

Polymer Type	Polymer Quantity (g)	Melting Point (°C)	Press Temperature (°C)	Time on Heat Press (min)
LDPE	1	135	150	10
PS	1	240	260	5
PLA	1.5	~150	200	4

## Data Availability

All the nucleotide sequences from the actinomycete strains were deposited in GenBank under accession numbers MT830750-MT830834, available at https://www.ncbi.nlm.nih.gov/genbank/ (accessed on 13 October 2022).

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
