# Peer review of "Marine-Derived Actinomycetes: Biodegradation of Plastics and Formation of PHA Bioplastics—A Circular Bioeconomy Approach"

_marinedrugs, 2022, doi:10.3390/md20120760_

Round 1

Reviewer 1 Report

Dear Authors

This is for sure an important topic, however, I think the focus should be put on the polymer activation which may trigger or allow degradation and not really on the conversion. This is due to the fact that less than 1% of polymer is degraded in about half a year ... so not very impressive. But, the mechanism behind is truely important; so the formation of new functional units on the polymersurface etc. Hence I suggest rephrasing this manuscript towards this.

Some comments:

keywords can be more selective, no doublications with title an abstract

From Line 100 onwards check italics style of organisms designations; correct those to be italics throughout!

Fig. 1; the clear zones are not well defined nor visible. Should be improved.

Fig. 3; is not clear and too small - need to be improved. ... same is true for Fig. 4.

The UV treatment need to be outlined in more detail, to allow reproducibility.

The transition from polymer degradation towards PHA formation is not well organized or structured. Indeed that is an interesting idea; but PHA formation on cheaper waste streams is much more efficient... hence biodegradation and biomass formation can be a result / goal already.

References need to be corrected; here also style need to checked, eg. of organisms and Journal abbreviations.

Supplemental Material needs proper legends

... final comment PHA was formed not produced as indicated by the title.

Author Response

The authors thank the reviewer for the detailed revision, comments and suggestions, which definitely improved our manuscript.

Reviewer 1:

  1. This is for sure an important topic, however, I think the focus should be put on the polymer activation which may trigger or allow degradation and not really on the conversion. This is due to the fact that less than 1% of polymer is degraded in about half a year ... so not very impressive. But, the mechanism behind is truely important; so the formation of new functional units on the polymer surface etc. Hence I suggest rephrasing this manuscript towards this.

The goal of this work was to evaluate the possibility of plastic waste (LDPE, PS, PLA which are persistent polluting plastics) being used by actinomycetes in the perspective of not only promoting its biodegradation but also assessing its potential valorization through the formation of biodegradable plastics, namely polyhydroxyalkanoates (PHAs), which are currently produced using other carbon sources.  Thus, the study was directed to achieve these answers.

There are very few studies on plastic biodegradation by actinomycetes [1], and studies that evaluate their ability to produce PHAs are even less [2-4]. The novelty of the current work is two join these two subjects. To the best of our knowledge, this is the first study performed with marine-derived actinomycetes for both evaluating plastic biodegradation and PHA formation. The authors agree that the reported plastic degradation rates are not very impressive. However, the use of enzymes from plastic biodegrading bacteria can improve biodegradation yields up to 90%. Thus, it is important to find new bacterial species that can biodegrade plastics as it is important to pave the way for future valorization of plastic waste, which is a major societal challenge, by its use as carbon source for the production of biodegradable plastics. Nevertheless, the authors also agree that the mechanisms behind plastic biodegradation by bacteria is important and that aspect was discussed in section 2.3 and 3 figures were added to the supplementary materials Fig.S8-S10. Lines 232-235 (Figure S8), Lines 258 to 263 (Figure S9) and Lines 315 to 319.

[1] Oliveira J, Belchior A, da Silva VD, Rotter A, Petrovski Ž, Almeida PL, Lourenço ND and Gaudêncio SP (2020) Marine Environmental Plastic Pollution: Mitigation by Microorganism Degradation and Recycling Valorization. Front. Mar. Sci. 7:567126. doi: 10.3389/fmars.2020.567126

[2] Valappil SP, Bucke ÆC, Roy ÆI (2007) Polyhydroxyalkanoates in Gram-positive bacteria: Insights from the genera Bacillus and Streptomyces. Antonie van Leeuwenhoek 91:1–17. doi: 10.1007/s10482-006-9095-5

[3] Matias F, Bonatto D, Padilla G, Rodrigues MFA, Henriques JAP (2009) Polyhydroxyalkanoates production by actinobacteria isolated from soil. Can. J. Microbiol. Vol. 55. doi: 10.1139/W09-029

[4] Inoue D, Suzuki Y, Uchida T, Morohoshi J, Sei K (2016) Polyhydroxyalkanoate production potential of heterotrophic bacteria in activated sludge. J. Biosci Bioeng. Vol. 121:1. doi: 10.1016/j.jbiosc.2015.04.022

  1. keywords can be more selective, no doublications with title an abstract

Thank you for your suggestion. Some of the keywords were deleted and/or changed. Lines 37 to 39.

  1. From Line 100 onwards check italics style of organisms designations; correct those to be italics throughout!

Thank you for noticing that. It has been corrected. Lines 107 to 109.

  1. Fig. 1; the clear zones are not well defined nor visible. Should be improved.

In Fig.1 the halos delineating white line has been deleted to improve their visualization. Line 115.

  1. Fig. 3; is not clear and too small - need to be improved. ... same is true for Fig. 4.

Fig.3 and Fig.4 were improved. Line 223 and line 264.

  1. The UV treatment need to be outlined in more detail, to allow reproducibility.

Information was added to section 3.4 to clarify the UV treatment procedure. It now reads “After the production of the polymeric thin plastic films, half of the LDPE, PS and PLA prepared films were treated using UV light at Wavelength λ = 254 nm (Elnorlux, 3907RVT815, 15 W, 230 V and 50 Hz) by direct exposure during 30 min, and stored at room temperature for further use” Lines720 to 723.

  1. The transition from polymer degradation towards PHA formation is not well organized or structured. Indeed that is an interesting idea; but PHA formation on cheaper waste streams is much more efficient... hence biodegradation and biomass formation can be a result / goal already.

The authors agree that PHA production using cheap waste streams other than plastic waste is much more efficient. However, the possibility of plastic waste being used by actinomycetes for PHA production was assessed in the perspective of not only promoting the production of a biodegradable plastic that can be efficiently obtained using different carbon sources, but also the valorization of plastic waste that is currently polluting the environment. Further discussion about PHA was added to the manuscript Lines 397 to 402, Lines 493 to 501, Lines 575 to 612, Lines 656 to 657.

  1. References need to be corrected; here also style need to checked, eg. of organisms and Journal abbreviations.

The references have been checked and the style changed in the revise manuscript file.

  1. Supplemental Material needs proper legends

All the figures and legends in the supplemental material have been changed for clarification and improved visualization.

  1. ... final comment PHA was formed not produced as indicated by the title.

The title has been changed and it now reads: “Marine-derived Actinomycetes: Biodegradation of Plastics and Formation of PHA Bioplastics—A Circular Bioeconomy Approach”. We have decided to change the name of the phylum Actinobacteria to the name of the class “Actinomycetes”, as this is a much commonly used name in the literature.

Reviewer 2 Report

The abstract states a high efficiency in the biodegradation of the plastics used, but the quantitative results do not agree with this statement. It is recommended to reconstruct the sentence.

The abstract is descriptive but does not report quantitative results. It is recommended to report the biodegradation yields and the changes that were analyzed by FTIR ATM in the possible degradation mechanism.

Figure 1 definitely does not show the halos reported by the authors, it is recommended to change the image to facilitate this appreciation. Additionally, the legend of this figure does not clarify some aspects such as: these photos were taken at what culture time, under what conditions, and was only one test per sample performed?

In the different trials, how was the biomass separated from the remaining plastic? 

Does the addition of detergent not prevent the actinobacteria from binding to the surface of the material to be biodegraded?

Were the plastics used in the tests free of plasticizing agents?

It is claimed that the inclusion of yeast extract in the medium improved degradative performance. But can this not be a consequence of having an alternate nitrogen and carbon source that leads to higher biomass and thus higher biodegradative activity? Was biomass quantified in all trials?.

Keeping up with biodegradation by FTIR ATM if it is the best strategy? even more so when the % biodegradation was so small? 

Since the identity of the strains is known, can't the genome of each strain be analyzed to know if it has the necessary machinery for PHA synthesis?

PHA accumulation was followed by nile blue, how specific is this method?, since in microphotographs the vacuoles are not evaluated but the whole cell is seen.

The authors say, "The different results of this study compared to the literature may be due to differences in culture media volume and acclimation of the strain to LDPE", what was the culture media volume, and what were the differences in acclimation of the strain to LDPE versus the other studies?

Author Response

The authors thank the reviewer for the detailed revision, comments and suggestions, which definitely improved our manuscript.

Reviewer 2:

  1. The abstract states a high efficiency in the biodegradation of the plastics used, but the quantitative results do not agree with this statement. It is recommended to reconstruct the sentence.

The whole abstract has been reconstructed and it now reads: “Thirty-six marine-derived actinobacteria strains were screened for their plastic biodegradability potential. Among these, Streptomyces gougerotti, Micromonospora matsumotoense, and Nocardiopsis prasina, revealed ability to degrade plastic films - low density polyethylene (LDPE), polystyrene (PS) and polylactic acid (PLA)- in varying conditions, namely upon the addition of yeast extract to the culture media and the use of UV pre-treated plastic films.”. Lines 23 to 28.

  1. The abstract is descriptive but does not report quantitative results. It is recommended to report the biodegradation yields and the changes that were analyzed by FTIR ATM in the possible degradation mechanism.

Biodegradation yields were added to the abstract, as well as the main changes analyzed by FTIR-ATR during biodegradation and mechanical tests. The following text was added: “S. gougerotti degraded 0.56 % of LDPE films treated with UV radiation and 0.67 % of PS films when inoculated with yeast extract. Additionally, N. prasina degraded 1.27 % of PLA films when these were treated with UV radiation, and yeast extract was added to the culture medium. The main and most frequent differences observed in FTIR-ATR spectra during biodegradation occurred at 1740 cm-1, indicating the formation of carbonyl groups and increase in the intensity of the bands, which indicates oxidation, Young Modulus decreased 30% in average.” Lines 28 to 34.

  1. Figure 1 definitely does not show the halos reported by the authors, it is recommended to change the image to facilitate this appreciation. Additionally, the legend of this figure does not clarify some aspects such as: these photos were taken at what culture time, under what conditions, and was only one test per sample performed?

In Fig.1 the halos delineating white line has been removed to improve their visualization. The requested information was added to the legend of Fig. 1. Both the biodegradation screening in agar plates and liquid media with the with plastics thin films were performed in triplicate, as well as their biodegradation evaluation. Lines 115 to 122. Moreover, this information was also added to the Methods and Experimental section. Lines 163, 682, 694, 744, 761, 769, and 785.

  1. In the different trials, how was the biomass separated from the remaining plastic? 

During the different trials, the plastic films were removed with the help of sterile forceps, in a sterilized flow chamber. Then, the formed bacterial biofilm in the plastic films surface was washed off with 2% (w/v) SDS and shaking for 5h, then washed with distilled water. The films were placed on a filter paper and dried overnight in a laboratory oven at 50 ºC. This information was included in section 3.6. Lines 747 to 750.

  1. Does the addition of detergent not prevent the actinobacteria from binding to the surface of the material to be biodegraded?

As a detergent, SDS most probably induces denaturation of the bacterial surface proteins that are involved in the processes of attachment and biofilm formation. SDS is able to inhibit biofilm formation and disperse mature biofilms of E. coli [1]. The minimum inhibitory concentration (MIC) and minimum biofilm inhibitory concentration (MBIC) for E. coli, were 1000 μg/mL and 64 μg/mL, respectively.

[1] Li L, Molin S, Yang L, Ndoni S (2013) Sodium Dodecyl Sulfate (SDS)-Loaded Nanoporous Polymer as Anti-Biofilm Surface Coating Material. Int J Mol Sci. 14(2), 3050-64. doi: 10.3390/ijms14023050

In our work, SDS was added in two situations:

(1) To prepare the solid media with emulsified polymers. In this case, 0.01 g of SDS were added to 50 ml final volume, which corresponds to a final concentration of 200 mg/ml.

(2) To remove the bacterial biofilm from the surface of the plastic thin films, in the biodegradation assays. Here, a 2% SDS solution was used, which corresponds to a final concentration of 20 000 mg/ml.

In the first situation (1) we worked with a concentration of SDS that inhibits biofilm formation, without killing the bacteria. Thus, in solid medium, the growth of actinomycetes was not inhibited, and so the capacity of these bacteria to produce and excrete enzymes that degrade the plastics was not compromised. In this experimental set-up, the bacteria are not expected to form a biofilm at the surface of the Petri dishes.

In contrast, in the second situation (2) our goal was to remove the bacterial biofilm formed at the surface of the thin plastic films, and we worked with a SDS concentration that was significantly above the MIC and the MBIC. After this, the films were rinsed with distilled water and allowed to dry overnight.

  1. Were the plastics used in the tests free of plasticizing agents?

The plastics used in the plastic biodegradation screening tests were all pure compounds. They do not incorporate plasticizing compounds such as phthalates.

  1. It is claimed that the inclusion of yeast extract in the medium improved degradative performance. But can this not be a consequence of having an alternate nitrogen and carbon source that leads to higher biomass and thus higher biodegradative activity? Was biomass quantified in all trials?

Yes, it can be a reason for faster biodegradation in short term. If the biomass quantity increases due to the presence of an alternate nitrogen and carbon source, plastic biodegradation can also improve, since there will be more cells available to degrade the same plastic amount. The biomass was not quantified in the biodegradation assays. In liquid media actinomycetes form round-shaped visible clusters/aggregates of different sizes, they do not form a turbid solution as other bacterial types, which makes OD600 quantification a very hard task perform. A higher number (about the double) of clusters/aggregates were observed when yeast extract was added.

  1. Keeping up with biodegradation by FTIR ATM if it is the best strategy? even more so when the % biodegradation was so small? 

There are alternative methods to measure and monitor biodegradation. Nevertheless, FTIR-ATR is one of the most used methods to determine structural chemical changes caused by microorganisms on the surface of plastic films. As the weight loss biodegradation % was small, using this sensitive technique allowed to reveal the chemical changes in the plastic thin films after biodegradation.

  1. Since the identity of the strains is known, can't the genome of each strain be analyzed to know if it has the necessary machinery for PHA synthesis?

Our strains were identified by 16S rRNA gene sequencing. We have not sequenced the genome of our strains. The genome sequence of strains of these species are not available, except for other species of the same genera. Those genomes are in a scaffold/contig format, thus annotation of the genome of our strains would be a laborious task. The work that we describe in this manuscript was the most effective way that we found to understand whether these actinomycetes would be able to form PHA. We are now optimizing these methods and the mentioned actinomycetes have been proved to be capable of producing PHA (through monomer identification by GC-MS), which will be published as soon as the all fermentation optimization assay will be concluded. In the future, genome sequencing and analysis will be performed to identify genes/operons involved in these processes.

  1. PHA accumulation was followed by nile blue, how specific is this method? since in microphotographs the vacuoles are not evaluated but the whole cell is seen.

Sudan black staining has been extensively used to detect the presence of lipophilic cellular inclusions like PHA, but it is prone to false positive errors due to staining of other lipid storage compounds. The use of the fluorescent dye Nile blue has revealed to be more specific and more sensitive than Sudan black for PHA granule imaging [1]. In fact, inclusions like glycogen or poly-P are not stained by Nile blue and its adsorption by cell walls and other lipid-containing cell components are apparently not sufficient for fluorescence do be detected. Thus, Nile blue staining is considered a rapid and direct screening test for PHA-positive cells.

[1] Koller M, Rodriguez-Contreras A (2015) Eng. Life Sci. 15, 558-581, https://onlinelibrary.wiley.com/doi/epdf/10.1002/elsc.201400228

  1. The authors say, "The different results of this study compared to the literature may be due to differences in culture media volume and acclimation of the strain to LDPE", what was the culture media volume, and what were the differences in acclimation of the strain to LDPE versus the other studies?

Our experiments were performed in 100 mL of culture media. In some studies [1,2], before the biodegradation assays, strains were acclimatized to the polymers. In our experiments, the strains used in the biodegradation assays were only pre-inoculated (20 mL) in the medium supplemented with yeast extract, without the polymers. These experimental volumes were added to the Methods and Experimental section.

[1] Denaro, R. et al (2020) Science of The Total Environment 749, 141608, https://doi.org/10.1016/j.scitotenv.2020.141608

[2] Skariyachan S. et al (2017) Environmental Science and Pollution Research  24, 8443–8457, https://doi.org/10.1007/s11356-017-8537-0

Reviewer 3 Report

This manuscript reports on the degradation of 3 types of bioplastics (LDPE, PS, PLA) coupling with the production of PHA by selected marine actinobacteria. The content of the manuscript fits well with the aim and scope of the journal. This interesting manuscript is well written with clear representation and should attract a wide readership. Only few comments listed below. Therefore, I recommend a minor revision of this manuscript.

Major points

1. Why the full-strength seawater is not used in cultivation of actinobacteria?

2. It is interesting to see that the PLA degradation is better with UV treatment. In my previous experiment, the UV treatment was found to make the PLA more difficult to degrade. Would you be able to explain this different observation?

3. Please provide the information on the cell concentration of each actinobacteria used in cfu/ml and what is the cultivation condition for inoculum preparation?

Minor points

Line 100, 101, 103 Italicize the name of actinobacteria

Line 355 Add reference for the claim of 300 PHA producing bacteria in this sentence.

Line 380-381 Revise the legend of Figure 8 Explain what is 8a and 8b

Line 396 8% weight loss?

Author Response

The authors thank the reviewer for the detailed revision, comments and suggestions, which definitely improved our manuscript.

Reviewer 3:

This manuscript reports on the degradation of 3 types of bioplastics (LDPE, PS, PLA) coupling with the production of PHA by selected marine actinobacteria. The content of the manuscript fits well with the aim and scope of the journal. This interesting manuscript is well written with clear representation and should attract a wide readership. Only few comments listed below. Therefore, I recommend a minor revision of this manuscript.

  1. Why the full-strength seawater is not used in cultivation of actinobacteria?

NaCl is often used to prevent bacterial growth (e.g. preservation by salty fish or meat). Thus, a more diluted concentration of seawater improves the actinomycete growth. It is described in the literature on conventional medium are more frequently pathogenic and that diluted medium favors bacteria which have more diverse metabolic functions, a characteristic of actinomycetes. [1]  In fact, one of the main parameters for yield optimization is often the NaCl concentration. [2-5] 

[1]  (https://link.springer.com/article/10.1007/s12275-019-9175-7) 

[2]  Meereboer, K.W.; Misra, M.; Mohanty, A.K. Review of recent advances in the biodegradability of polyhydroxyalkanoate (PHA) bioplastics and their composites. Green Chem. 2020, 22, 5519–5558, doi:10.1039/d0gc01647k.

[3]   PRATT, D.B.; WADDELL, G. Adaptation of marine bacteria to growth in media lacking sodium chloride. Nature 1959, 183, 1208–1209, doi:https://doi.org/10.1038/1831208a0.

[4]  Fernandez-Linares, L.; Acquaviva, M.; Bertrand, J.-C.; Gauthier, M. Effect of sodium chloride concentration on growth and degradation of eicosane by the marine halotolerant bacterium Marinobacter hydrocarbonoclasticus. Syst. Appl. Microbiol. 1996, 19, 113–121, doi:https://doi.org/10.1016/S0723-2020(96)80018-X.

[5]  Said, G.; Ahmad, F. Effects of salt concentration on the production of cytotoxic geodin from marine-derived fungus Aspergillus sp. Turkish J. Biochem. 2022, 47, 399–402, doi:10.1515/tjb-2022-0058.100. Jarerat, A.; Pranamuda, H.; Tokiwa, Y. Poly(L-lactide)-degrading activity in various actinomycete. Macromol. Biosci. 2002, 2, 420–428, doi:10.1002/MABI.200290001.

  1. It is interesting to see that the PLA degradation is better with UV treatment. In my previous experiment, the UV treatment was found to make the PLA more difficult to degrade. Would you be able to explain this different observation?

In our case, the UV treatment improved the biodegradation of PLA films. Considering the literature and our results, the UV radiation treatment may have caused the break of the PLA ester linkage, resulting in the generation of photooxidized products. This could be advantageous to increase the hydrophilic property of PLA, which might contribute to a greater accessibility of the polymer [1,2].

[1] Koo GH, Jang J (2008) Surface Modification of Poly(Lactic Acid) by UV/Ozone Irradiation. Fibers and Polymers, Vol.9, No.6, 674-678. doi: 10.1007/s12221-008-0106-1

[2] Zhang C, Goddard J, Constantine K, Collins P (2013) The Effect of UV Treatment on the Degradation of Compostable Polylactic Acid. J Emerg Invest ISSN 2638-0870)

  1. Please provide the information on the cell concentration of each actinobacteria used in cfu/ml and what is the cultivation condition for inoculum preparation?

The strains used in LDPE, PS and PLA biodegradation assays were grown in SSW media (20 mL) with 0.1 % (m/v) of yeast extract. In liquid media actinomycetes form round-shaped visible clusters/aggregates of different sizes, they do not form a turbid solution as other bacterial types, which makes OD600 quantification hard to perform. A higher number of clusters/aggregates was observed when yeast extract was added (about the double). In all the biodegradation assays approximately 10 % of the strains clusters/aggregates (biomass) were transferred with a sterilized volumetric pipet to fresh SSW media (100 mL) and inoculated with the thin plastic films. This information was clarified in section 3.5, Lines 733 and 745.

  1. Line 100, 101, 103 Italicize the name of actinobacteria

Thank you for noticing that. It has been corrected.

  1. Line 355 Add reference for the claim of 300 PHA producing bacteria in this sentence.

Reference [67] was added to the sentence to support the claim of 300 PHA producing bacteria.

  1. Line 380-381 Revise the legend of Figure 8 Explain what is 8a and 8b

The legend has been revised. Lines 424 to 426.

  1. Line 396 8% weight loss?

The authors thank the reviewer for noticing that. In fact, it is 8 % weight loss. The word “loss” was added to the sentence. Line 442.

Reviewer 4 Report

The work isolated microbials from the ocean and applied them to the microbial remediation of plastic pollution. The bacteria used have shown great potential in biodegrading LDPE, PS, and PLA thin plastic films, and the formation of biodegradable plastic material PHA. This manuscript was prepared well.  I think it is appropriate to be published in the journal of Marine Drugs with minor revisions.

Figure 1: How long did it take for the three actinobacteria to form a clear zone halo on the plastic emulsified media?

Line 110. Why did the authors choose thin plastic films as the degradation substrate after showing that the three bacteria could degrade conventional plastics? One would assume that the degradation of conventional plastics could more important since they are more common in real life.

Line 112-113. Also, please explain the reason why specific bacteria were subjected to the degradation of the very type of plastic films.

Line 182. There is no need to explain FTIR-ATR in the section of Result and Discussion. It can be rearranged to the section of Method.

Figures 3 and 4 are too blurry to read, pictures with higher quality are required. Both the x-axis and y axes should be labeled accurately and clearly.

Figure 5. The red arrows that possibly indicated the difference brought by the supplemented with yeast extract should be labeled more clearly. An inset on the raw spectrum is recommended.

Line 477. A further discussion on the mechanism of the yeast extract as an important co-factor for the biodegradation of PS by multiple bacteria is recommended.

Line 518. Since the Discussion section was not individually set in this manuscript, direct experimental evidence should be indicated in this sentence to make a claim. The author only cited a paper, instead of comparing their data of the observation of PHA produced by actinobacteria.

Line 526. The further development of producing PHA by using actinobacteria should be discussed in more detail if the authors intended to keep this part in the manuscript.

Author Response

Reviewer 4:

The work isolated microbials from the ocean and applied them to the microbial remediation of plastic pollution. The bacteria used have shown great potential in biodegrading LDPE, PS, and PLA thin plastic films, and the formation of biodegradable plastic material PHA. This manuscript was prepared well.  I think it is appropriate to be published in the journal of Marine Drugs with minor revisions.

  1. 1. Figure 1: How long did it take for the three actinobacteria to form a clear zone halo on the plastic emulsified media?

Streptomyces gougerotti took 17 days; Micromonospora matsumotoense took 14 days; and Nocardiopsis prasine took 7 days. This information was added to the legend of Figure 1. Lines 116 to 121.

  1. Line 110. Why did the authors choose thin plastic films as the degradation substrate after showing that the three bacteria could degrade conventional plastics? One would assume that the degradation of conventional plastics could more important since they are more common in real life.

The polymeric materials used as degradation test both in Petri dishes and as thin films are conventional plastics (LDPE, PS and PLA). We used several thermoplastics that are widely used in common applications such as plastic bottles. The plastics were shaped into thin films in order to uniformize the shape of the plastic pieces that can be found in nature, and to allow an easier assessment of the degradation at the film’s surface both by weight-loss, FTIR-ATR and mechanical tests. This information was clarified in the manuscript. Lines 126 to 129.

  1. Line 112-113. Also, please explain the reason why specific bacteria were subjected to the degradation of the very type of plastic films.

The selection of the specific bacteria and of the type of plastic film to be used in each degradation assay was performed based on the results of the screening biodegradation tests for the 36 actinomycete strains previously performed in plastic emulsified Petri dishes. This information was clarified in the text. Lines Lines 102 to 108, Lines 123 to 125, and Lines 680 to 687.

  1. Line 182. There is no need to explain FTIR-ATR in the section of Result and Discussion. It can be rearranged to the section of Method.

Thank you for the suggestion. We have deleted the phrase since it was already mentioned in the Methods section.

  1. Figures 3 and 4 are too blurry to read, pictures with higher quality are required. Both the x-axis and y axes should be labeled accurately and clearly.

The images have now higher quality. The y axis is "Transmittance (%) and the x axis is Wave Number (cm-1)". The spectra were inserted in the manuscript as they were obtained from the FTIR-ATR equipment. Nevertheless, this information was added to the figures legends and to the experimental section. Lines 226 to 227, Lines 267 to 268, Lines 296 to 297, and Line 767 to 768.

  1. Figure 5. The red arrows that possibly indicated the difference brought by the supplemented with yeast extract should be labeled more clearly. An inset on the raw spectrum is recommended.

The image has been changed according to the suggestions and the information was added to the caption.

  1. Line 477. A further discussion on the mechanism of the yeast extract as an important co-factor for the biodegradation of PS by multiple bacteria is recommended.

The sentence has been changed to “Yeast extract was added to the media to promote the growth of actinomycetes biomass”. As the word “co-factor” could lead to a misinterpretation. Line 184 to 186.

  1. Line 518. Since the Discussion section was not individually set in this manuscript, direct experimental evidence should be indicated in this sentence to make a claim. The author only cited a paper, instead of comparing their data of the observation of PHA produced by actinobacteria.

As suggested, we included direct experimental evidence to compare our data with literature. Lines 397 to 402, Lines 493 to 501, Lines 575 to 612, Lines 656 to 657.

  1. Line 526. The further development of producing PHA by using actinobacteria should be discussed in more detail if the authors intended to keep this part in the manuscript.

As suggested, PHA production from plastic waste using actinomycetes was discussed in more detail. Lines 397 to 402, Lines 493 to 501, Lines 575 to 612, Lines 656 to 657.

Round 2

Reviewer 1 Report

all my requests have been changed; still I think it is more an idea from polymer to polymer.